# Noradrenaline release from the locus coeruleus shapes stress-induced hippocampal gene expression

Mattia Privitera[1,2], Lukas M von Ziegler[1,2], Amalia Floriou-Servou[1,2], Sian N Duss[1,2], Runzhong Zhang[1], Rebecca Waag[1,2], Sebastian Leimbacher[1], Oliver Sturman[1,2], Fabienne K Roessler[1], Annelies Heylen[3], Yannick Vermeiren[3,4], Debby Van Dam[3,5], Peter P De Deyn[3,5,6], Pierre-Luc Germain[1,2,7,8], Johannes Bohacek[1,2]*

[1]Laboratory of Molecular and Behavioral Neuroscience, Institute for Neuroscience, Department of Health Sciences and Technology, ETH Zurich, Zurich, Switzerland; [2]Neuroscience Center Zurich, ETH Zurich and University of Zurich, Switzerland, Zurich, Switzerland; [3]Laboratory of Neurochemistry and Behavior, Experimental Neurobiology Unit, Department of Biomedical Sciences, University of Antwerp, Antwerp, Belgium; [4]Division of Human Nutrition and Health, Chair Group of Nutritional Biology, Wageningen University & Research (WUR), Wageningen, Netherlands; [5]Department of Neurology and Alzheimer Center, University of Groningen and University Medical Center Groningen (UMCG), Groningen, Netherlands; [6]Department of Neurology, Memory Clinic of Hospital Network Antwerp (ZNA) Middelheim and Hoge Beuken, Antwerp, Belgium; [7]Computational Neurogenomics, Institute for Neuroscience, Department of Health Sciences and Technology, ETH Zürich, Zurich, Switzerland; [8]Laboratory of Statistical Bioinformatics, University of Zürich, Zürich, Switzerland

*For correspondence:
johannes.bohacek@hest.ethz.ch

Competing interest: The authors declare that no competing interests exist.

**Abstract** Exposure to an acute stressor triggers a complex cascade of neurochemical events in the brain. However, deciphering their individual impact on stress-induced molecular changes remains a major challenge. Here, we combine RNA sequencing with selective pharmacological, chemogenetic, and optogenetic manipulations to isolate the contribution of the locus coeruleus-noradrenaline (LC-NA) system to the acute stress response in mice. We reveal that NA release during stress exposure regulates a large and reproducible set of genes in the dorsal and ventral hippocampus via β-adrenergic receptors. For a smaller subset of these genes, we show that NA release triggered by LC stimulation is sufficient to mimic the stress-induced transcriptional response. We observe these effects in both sexes, and independent of the pattern and frequency of LC activation. Using a retrograde optogenetic approach, we demonstrate that hippocampus-projecting LC neurons directly regulate hippocampal gene expression. Overall, a highly selective set of astrocyte-enriched genes emerges as key targets of LC-NA activation, most prominently several subunits of protein phosphatase 1 (*Ppp1r3c*, *Ppp1r3d*, *Ppp1r3g*) and type II iodothyronine deiodinase (*Dio2*). These results highlight the importance of astrocytic energy metabolism and thyroid hormone signaling in LC-mediated hippocampal function and offer new molecular targets for understanding how NA impacts brain function in health and disease.

## eLife assessment

This **important** paper uses a multifaceted approach to implicate the locus coeruleus-noradrenaline system in the stress-induced transcriptional changes of dorsal and ventral hippocampus. It provides

an inventory of dorsal and ventral hippocampal gene expression upregulated by activation of LC-NA system, which can be used as starting point for more functional studies related to the effects of stress-induced physiological and pathological changes. The results **convincingly** support the conclusions. This paper will be of interest to those interested in stress neurobiology, hippocampal, and/or noradrenaline function.

## Introduction

When an organism faces an acutely stressful situation, a set of evolutionarily conserved mechanisms are triggered to re-route all available resources to body functions that enhance performance and increase the chance of survival (*Floriou-Servou et al., 2021*; *Joëls and Baram, 2009*). In the brain, stress exposure immediately activates the locus coeruleus-noradrenaline (LC-NA) system. Although the LC is a heterogeneous structure with modular organization, it appears that in stressful situations, broad activation of the LC - and subsequent widespread NA release throughout the brain - serves as a broadcast signal to orchestrate re-routing of computational resources to meet situational demands (*Likhtik and Johansen, 2019*; *Poe et al., 2020*). On the network level, for example, NA release from the LC is sufficient to trigger a rapid reconfiguration of large-scale networks that shift processing capacity toward salience processing (*Zerbi et al., 2019*; *Oyarzabal et al., 2022*). On a circuit level, forebrain regions seem to be particularly important targets of the LC-NA system to influence cognitive processes and ultimately behavior. This involves the engagement of anxiety and memory circuits including the amygdala, hippocampus, and prefrontal cortex, which leads to an increase in avoidance behavior (*McCall et al., 2015*; *McCall et al., 2017*; *Hirschberg et al., 2017*; *Zerbi et al., 2019*) and supports memory formation of salient events (*Uematsu et al., 2015*; *Hansen, 2017*; *Sara, 2015*; *Schwabe et al., 2022*).

While the insights into network and circuit dynamics of the LC have been galvanized by recent advances in circuit neuroscience tools, our understanding of the molecular impact of NA release has remained largely unexplored. However, we know that the stress response triggers multifaceted molecular cascades that profoundly change brain function and behavior in response to stressful events (*Floriou-Servou et al., 2021*; *Joëls and Baram, 2009*). These molecular changes are mediated by a large number of neurotransmitters, neuropeptides, and hormones, which operate on different time scales, to allow rapid activation, sustained activity, and successful termination of the stress response. Many of the rapid molecular changes induced by stress exposure seem to be driven by NA. For example, several of the genes that are induced by an acute stress challenge can be blocked by β-adrenergic receptor antagonists (*Roszkowski et al., 2016*). However, a similar analysis on the genome-wide level has not been conducted. Further, it remains unknown whether NA release alone is sufficient to trigger transcriptomic changes in the first place. Here, we profile the molecular fingerprint of stress-induced NA release by combining pharmacological, chemogenetic, and optogenetic manipulation of the LC-NA system with genome-wide transcriptomic analyses. We focus on the hippocampus, as it receives its sole NA supply from the LC (*Loy et al., 1980*; *Robertson et al., 2013*; *Oleskevich et al., 1989*), and because the molecular response to acute stress has been characterized in great detail in this region (*von Ziegler et al., 2022*; *Mifsud et al., 2021*; *Floriou-Servou et al., 2018*). As the dorsal hippocampus (dHC) and ventral hippocampus (vHC) are involved in different circuits (*Strange et al., 2014*) and are transcriptionally very distinct (*Cembrowski et al., 2016*; *Floriou-Servou et al., 2018*), we analyze these regions separately.

## Results

A recent multi-omic characterization of the acute stress response in the mouse hippocampus revealed that stress-induced transcriptional changes peak between 45 and 90 min after stress exposure, before gradually returning to baseline (*von Ziegler et al., 2022*). To determine how NA might contribute to these effects, we first measured the dynamics of NA turnover in response to a cold swim stress exposure in the brain. Using ultra-high performance liquid chromatography (uHPLC), we determined concentrations of NA and its main metabolite 3-methoxy-4-hydroxyphenylglycol (MHPG) at various time points over 3 hr after swim stress exposure in the cortex (*Figure 1a*). NA turnover (as measured by the MHPG/NA ratio) peaked at 45 min and returned to baseline within 90 min after stress onset

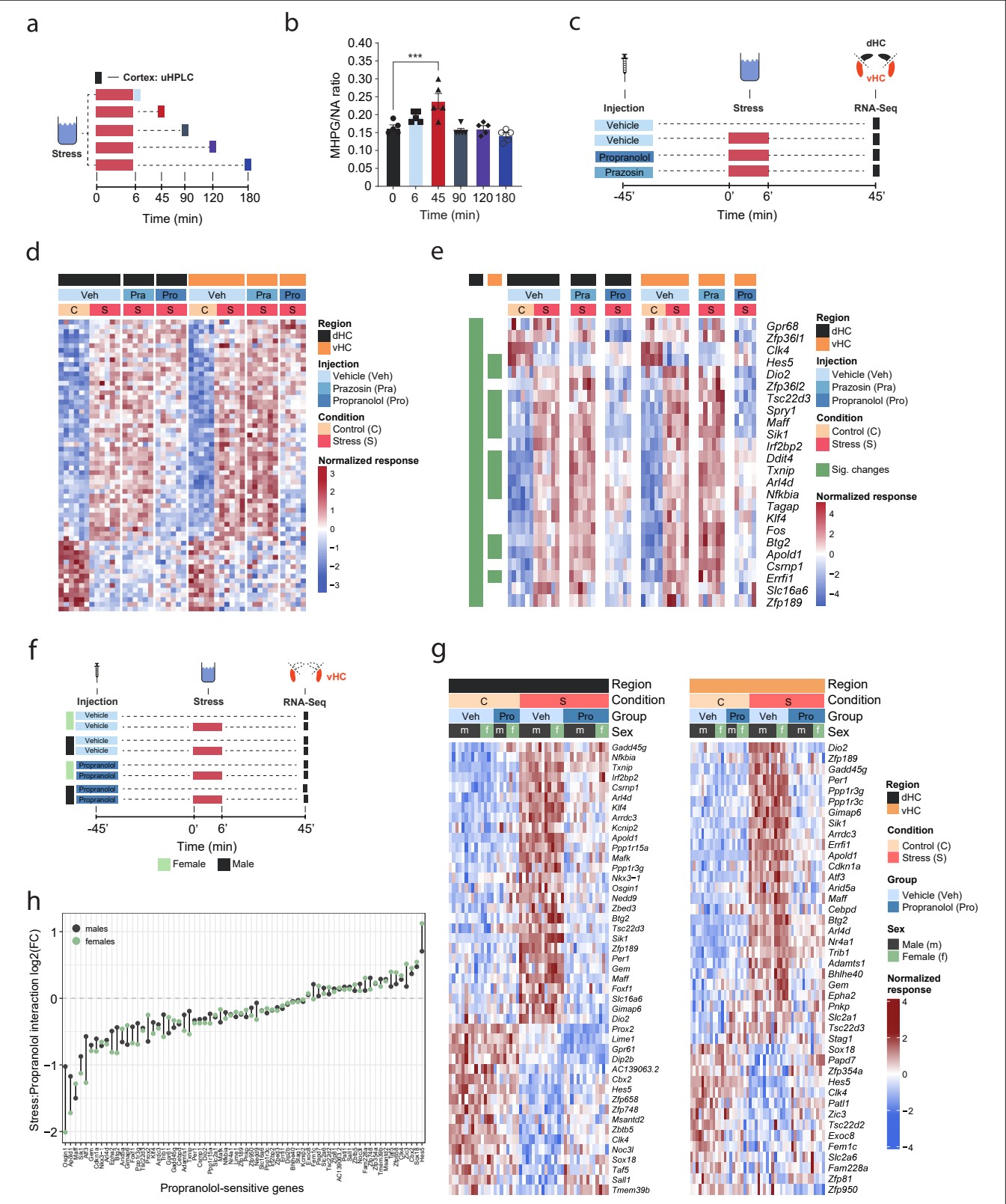

**Figure 1.** β-Adrenergic receptors mediate stress-induced transcriptomic changes in the hippocampus and are independent of subregion and sex. (**a**) Experimental design for assessing stress-induced cortical noradrenaline (NA) turnover at various time points following stress exposure. (**b**) Stress-dependent changes in cortical NA turnover, as measured by the ratio of 3-methoxy-4-hydroxyphenylglycol (MHPG) and NA levels. NA turnover significantly increased within 45 min and returned to baseline within 90 min of stress onset (one-way ANOVA with Tukey's post hoc tests; F(5, 24)=10.55,

*Figure 1 continued on next page*

Figure 1 continued

p<0.0001). n = 5 per group and individual data points are shown with bars representing mean ± s.e.m. (**c**) Experimental design for assessing the effect of prazosin (Pra, 1 mg/kg, i.p.) and propranolol (Pro, 10 mg/kg, i.p.) on stress-dependent transcriptomic changes in the dorsal (dHC) and ventral (vHC) hippocampus 45 min after stress exposure. (**d**) Heatmap showing all differentially expressed genes across dHC and vHC and pharmacological treatments 45 min after acute swim stress exposure. n = 6 per group. (**e**) Heatmap selectively showing those stress-responsive genes that are affected by the β-adrenergic receptor antagonist propranolol 45 min after acute swim stress exposure in the dHC and vHC (false discovery rate [FDR]-adjusted p<0.05). (**f**) Experimental design for assessing propranolol-dependent changes in the vHC of female and male mice. n = 5 per group. (**g**) Heatmap showing - across the two experiments - the expression of all stress-dependent genes that are affected by propranolol treatment in either region (dHC and vHC; FDR-adjusted p<0.05). (**h**) Comparison of the modulating effect of propranolol on the stress response (log2-fold-changes of the stress:propranolol interaction term) in males and females, across both experiments and regions.

The online version of this article includes the following figure supplement(s) for figure 1:

**Figure supplement 1.** In-depth analysis of noradrenergic contribution to stress-induced transcriptomic changes in the hippocampus.

(*Figure 1b*). Therefore, we chose the 45 min time point to assess the contribution of NA signaling to stress-induced transcriptomic changes. To this end, we blocked adrenergic receptors using either the α1-adrenergic receptor antagonist prazosin or the β-adrenergic receptor antagonist propranolol prior to stress exposure (*Figure 1c*). In line with our previous work (*von Ziegler et al., 2022*; *Floriou-Servou et al., 2018*; *Roszkowski et al., 2016*), acute swim stress induced a robust transcriptional response 45 min after stress exposure in both dHC and vHC (see Veh-C vs Veh-S in *Figure 1d*, *Figure 1—figure supplement 1a*). Prazosin only mildly impacted stress-dependent transcriptional changes in the dHC and vHC (see Veh-S vs Pra-S in *Figure 1d*, *Figure 1—figure supplement 1a*). Surprisingly, in stress-exposed animals prazosin seemed to slightly amplify - rather than prevent - some stress effects (*Figure 1—figure supplement 1b*). In contrast, propranolol had a profound impact, preventing many of the stress-induced changes in the dHC and vHC (see Veh-S vs Pro-S in *Figure 1d*, *Figure 1—figure supplement 1a*). Indeed, a direct comparison between the stress group and the stress+propranolol group found many genes that were significantly altered by propranolol administration (*Figure 1e*, *Figure 1—figure supplement 1c*). This response to propranolol was very similar in the dHC and vHC (*Figure 1e*).

While blocking β-adrenergic receptors was able to block stress-induced gene expression, we did not test whether propranolol might decrease gene expression already at baseline, independent of stress. Additionally, all tests had thus far been conducted in male mice, raising the question about potential sex differences in NA-mediated transcriptomic responses. To address these two issues, we repeated the experiment in both sexes and included a group that received a propranolol injection but was not exposed to stress (*Figure 1f*). Combining the data from both experiments, we repeated the analysis for each region, to identify genes whose response to stress was inhibited by propranolol (*Figure 1g*). As in the previous experiment, we found that many of the stress-induced gene expression changes were blocked by propranolol injection in both dHC (*Figure 1g*, left panel) and vHC (*Figure 1g*, right panel). Importantly, propranolol did not change the expression level of these genes in the absence of stress. We then directly compared the genes sensitive to stress and propranolol treatment in both dHC and vHC. To this end, we plotted the union of genes showing a significant stress:propranolol interaction in either region in one heatmap across both dHC and vHC (*Figure 1—figure supplement 1d*). This showed again that the stress-induced changes were very similar in dHC and vHC, and that propranolol similarly blocked many of them. Finally, we asked whether the response differs between males and females. Despite clear sex differences in gene expression at baseline (data not shown), we found no significant sex differences in response to stress or propranolol between male and female mice (false discovery rate [FDR] <0.05; *Figure 1g*). To more directly visualize this, we compared females and males by plotting the log2-fold-changes of the stress:propranolol interaction across all stress-induced genes that were blocked by propranolol. We find very similar regulation patterns in both sexes (*Figure 1h*). Although none of these sex differences are significant, some genes seem to show quantitative differences, so we plotted the expression patterns of the five genes showing the largest difference in interaction term as boxplots, which suggest that these spurious differences are likely due to noisy coefficient estimates (*Figure 1—figure supplement 1e*). To address concerns that our analysis of sex differences might not have been sufficiently powered, we performed a meta-analysis of the experiments shown here along with previously published datasets from our lab (*Floriou-Servou et al., 2018*; *von Ziegler et al., 2022*). In all these experiments, the vHC of male and

female mice was profiled 45 min after exposure to an acute swim stress challenge. This resulted in a sample size of 51 males and 20 females. Despite this high number of independent samples, we could not identify any statistically significant interaction between sex and the stress response. To identify candidates that might not reach significance while discounting differences due to noise in fold-change estimates, we reproduced the same analysis using DESeq2 with approximate posterior estimation for generalized linear model (apeglm) logFC shrinkage (*Zhu et al., 2019*). This analysis also did not reveal any sex differences in the stress response (*Figure 1—figure supplement 1f*). We then tailored the meta-analysis specifically to the set of stress-responsive genes that were blocked by propranolol, and also for these genes the response to stress was strikingly similar in both sexes (*Figure 1—figure supplement 1g*). Altogether, we conclude that there are no major sex differences in the rapid transcriptomic stress response in the hippocampus, and that blocking β-receptors prevents a large set of stress-induced genes in both females and males.

We then tested whether direct activation of hippocampal β-adrenergic receptors is sufficient to induce gene expression changes. To this end, we infused animals with the β-adrenergic receptor agonist isoproterenol into the dHC (because it is easier to target than the vHC). We then manually selected a few genes whose stress-induced induction was blocked by propranolol pre-treatment either partially (*Apold1*) or completely (*Dio2*, *Sik1*, and *Ppp1r3c*; see *Figure 1g*), and assessed the expression of these genes by targeted qRT-PCR assays (*Figure 1—figure supplement 1h*). Isoproterenol directly increased hippocampal expression of *Apold1*, *Dio2*, and *Sik1,* but not *Ppp1r3c* (*Figure 1—figure supplement 1i*). In summary, the results presented here show that for a large number of genes, NA signaling via β-adrenergic receptors is required to regulate the stress-induced transcriptional response.

An acute stress exposure triggers the release of a plenitude of stress mediators - neurotransmitters, peptides, and hormones - that interact to regulate molecular changes (*Joëls and Baram, 2009*). As it is unclear how NA interacts with other stress mediators, we asked whether we could isolate the molecular changes for which NA release is not only required, but sufficient, by triggering NA release in the hippocampus. Because hippocampal NA supply is provided exclusively by long-range projections from the LC (*Loy et al., 1980*; *Robertson et al., 2013*; *Oleskevich et al., 1989*), we first pharmacologically activated NA release using the α2-adrenergic receptor antagonist yohimbine, which strongly disinhibits noradrenergic neurons (*Figure 2a*). As expected, systemic administration of yohimbine (3 mg/kg, i.p.) led to a strong increase in cFos expression within the LC (*Figure 2b*) and increased NA turnover in the cortex (*Figure 2—figure supplement 1a*). Because NA mediates the stress-induced increase in anxiety (*McCall et al., 2015*; *Zerbi et al., 2019*), we also evaluated behavioral changes in the open field test. We had shown previously that acute stress increases anxiety in the open field test (*Sturman et al., 2018*; *von Ziegler et al., 2022*; *Figure 2—figure supplement 2a*), and very similarly yohimbine also suppressed locomotion and exploratory behaviors (*Figure 2—figure supplement 2b*). To directly compare the impact of stress and yohimbine injection on the transcriptomic response in the hippocampus, we then exposed mice to acute swim stress, or to yohimbine injection without stress exposure (*Figure 2c*). Yohimbine induced a clear and consistent transcriptional response in the dHC and vHC. Direct comparison between stress and yohimbine revealed no significant differences (*Figure 2d–e*, *Figure 2—figure supplement 1b*), suggesting that yohimbine administration alone can mimic a large fraction of the stress-induced transcriptional response. To more specifically probe whether selective activation of the LC-NA system alone is sufficient to trigger the observed changes in gene expression, we turned to direct activation of the LC. First, we used the chemogenetic actuator hM3Dq (*Zhu et al., 2016*) to trigger a strong and prolonged activation of the LC (*Figure 2f*). In support of previous work (*Zerbi et al., 2019*; *Privitera et al., 2020*), injection of an ultra-low dose of the potent hM3Dq-actuator clozapine (0.03 mg/kg) led to a strong cFos increase in tyrosine hydroxylase (TH) positive LC neurons in hM3Dq+, but not in hM3Dq- animals (*Figure 2g*). Further, we used fiber photometry combined with genetically encoded NA sensors to measure NA release in real time directly in the hippocampus (*Kagiampaki et al., 2023*; *Grimm et al., 2022*; *Feng et al., 2019*). We found that chemogenetic LC activation leads to a rapid and sustained increase in hippocampal NA in both females and males (*Figure 2—figure supplement 1c–e*). Behaviorally, chemogenetic LC activation induced an anxiety-like phenotype in the open field test, similar to the response to yohimbine (*Figure 2—figure supplement 2c*). Transcriptomic analysis revealed that chemogenetic LC activation induced significant transcriptomic changes that were similar in the dHC and vHC (*Figure 2h*). Overall,

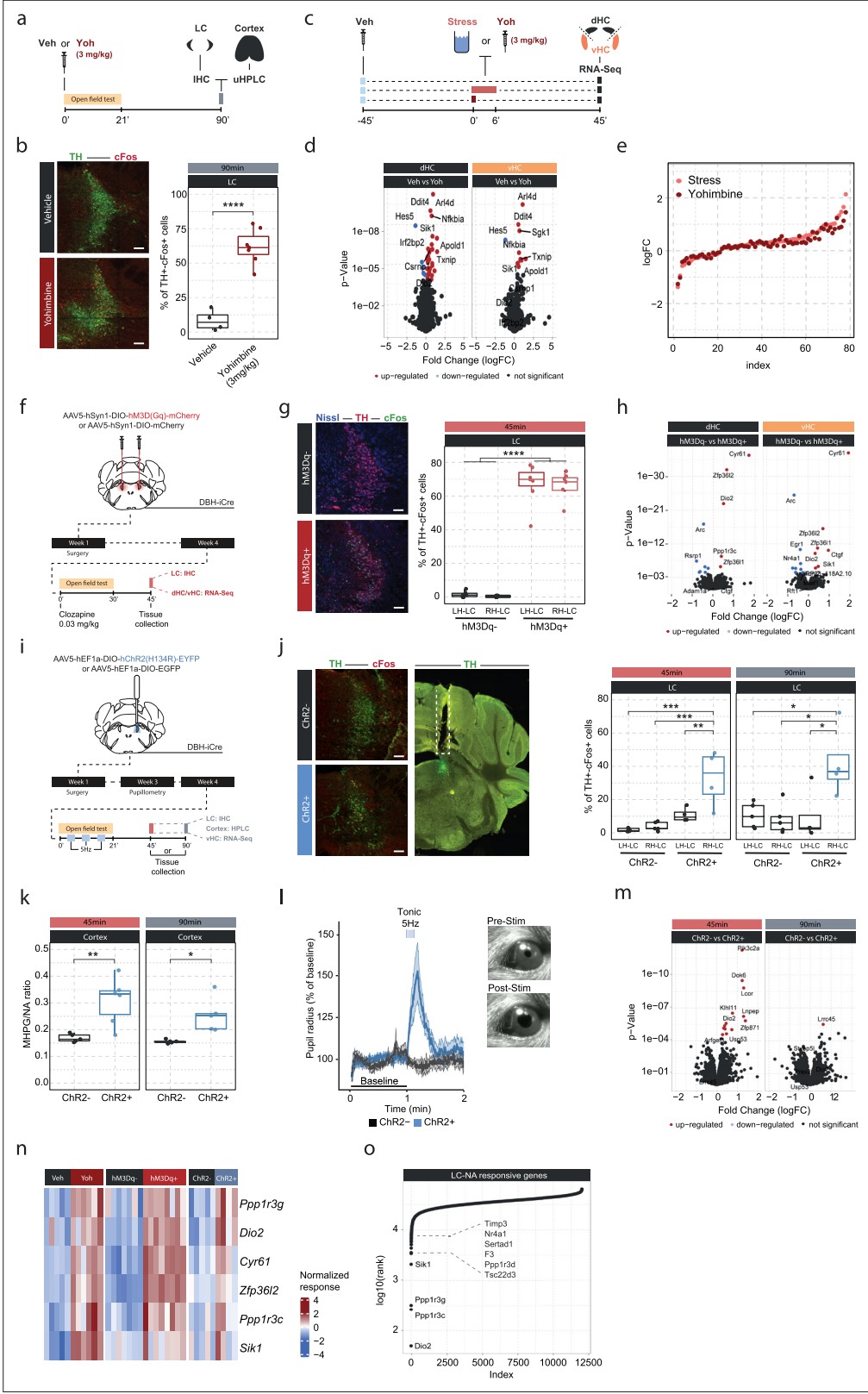

**Figure 2.** Locus coeruleus (LC)-mediated transcriptomic changes in the hippocampus. (**a**) Experimental design for assessing LC activation and cortical noradrenaline (NA) release induced by injection of yohimbine (3 mg/kg, i.p.). (**b**) Representative images and quantification of LC activation in mice 90 min after injection of vehicle or yohimbine as measured by cFos (red) and tyrosine hydroxylase (TH, green) coexpression within LC neurons.

*Figure 2 continued on next page*

*Figure 2 continued*

Yohimbine injection increased cFos expression within LC neurons compared to vehicle-injected animals (unpaired t-test; t(8.9)=−8.814, p=1.083e-05). Vehicle, n=4; yohimbine n=7. Scale bar: 100 µm. (**c**) Experimental design for comparing molecular changes in the hippocampus after acute swim stress exposure or yohimbine administration. (**d**) Volcano plots showing differentially expressed RNA transcripts in the dorsal (dHC) and ventral (vHC) hippocampus between control (Veh) and yohimbine (Yoh) injected animals 45 min after injection. Red and blue values represent changes with false discovery rate [FDR]-adjusted p<0.05 (Veh n = 6, Yoh n = 6). (**e**) Strength of the yohimbine effect in comparison to the transcriptomic stress response. Data are sorted by interaction strength in the stress group (orange) and the corresponding interaction strength of the yohimbine group are shown in dark red for the same gene. (**f**) Experimental design for assessing LC activation and hippocampal transcriptomic changes induced by chemogenetic LC activation. (**g**) Representative images and quantification of LC activation in mice 45 min after injection of clozapine (0.03 mg/kg) in hM3Dq- and hM3Dq+ animals as measured by cFos (green) and TH (red) coexpression within LC neurons. Neurons are stained with Nissl (blue). Clozapine injection increased cFos expression within LC neurons in hM3Dq+ animals compared to hM3Dq- animals (one-way ANOVA; F(3, 23)=135.4, p=9.34e-15). hM3Dq- n = 6, hM3Dq+ n = 7. Scale bar: 100 µm. (**h**) Volcano plots showing differentially expressed RNA transcripts between hM3Dq- and hM3Dq+ animals 45 min after injection of clozapine (0.03 mg/kg) in the dHC and vHC. Red and blue values represent changes with FDR-adjusted p<0.05 (hM3Dq- n = 6, hM3Dq+ n = 7). (**i**) Experimental design for assessing LC activation, cortical NA release, pupillometry and hippocampal transcriptomic changes induced by optogenetic 5 Hz LC activation. (**j**) Representative images and quantification of LC activation in mice after 5 Hz optogenetic LC activation as measured by cFos (red) and TH (green) coexpression within LC neurons in stimulated and non-stimulated LC hemispheres of ChR2- and ChR2+ animals. 5 Hz stimulation increased cFos expression within LC neurons in stimulated LC hemispheres of ChR2+, but not in ChR2- animals 45 min (one-way ANOVA with Tukey's post hoc tests; F(3, 14)=12.91, p=0.000256) and 90 min after stimulation onset (one-way ANOVA with Tukey's post hoc tests; F(3, 14)=5.866, p=0.00824). ChR2- (45 min), n=5; ChR2- (90 min), n=5; ChR2+ (45 min), n=4; ChR2+ (90 min), n=4. Scale bar: 100 µm. (**k**) Quantification of cortical 3-methoxy-4-hydroxyphenylglycol (MHPG)/NA ratio, as measured by ultra-high performance liquid chromatography (uHPLC), after 5 Hz optogenetic LC activation in ChR2- and ChR2+ animals. 5 Hz stimulation increased cortical NA turnover in ChR2+ animals (unpaired t-test; 45 min: t(3.6)=8.444, p=0.001681; 90 min: t(4.0854)=3.4127, p=0.02608). ChR2- 45 min, n=5; ChR2- 90 min, n=5; ChR2+ 45 min, n=6; ChR2+ 90 min, n=5. (**l**) Average pupil size changes in response to 5 Hz optogenetic LC activation in ChR2- and ChR2+ animals. 5 Hz stimulation increased pupil size in ChR2+, but not ChR2- animals. (**m**) Volcano plots showing differentially expressed RNA transcripts between ChR2- and ChR2+ animals 45 and 90 min after 5 Hz optogenetic LC activation in the vHC. Red and blue values represent changes with FDR-adjusted p<0.05 (ChR2- n = 10, ChR2+ n = 11). (**n**) Heatmap showing genes that are commonly differentially expressed by yohimbine, chemogenetic, and optogenetic-induced LC activation. (**o**) Logarithmic cumulative rank of genes across all experiments from *Figure 1* and Figure 2 in terms of their NA responsiveness. A lower cumulative rank indicates that a gene is among the more significant hits across all analyses (for full list of included analyses, see Methods). Labels indicate the 10 genes identified to be most responsive to LC-NA stimulation. *p<0.05, **p<0.01, ***p<0.001, ****p<0.0001. For all boxplots in this manuscipt, the bottom and top hinges represent the 1st and 3rd quartiles, while the whiskers extends from the box hinge to the farthest value no further than +/-1.5*IQR from the hinge.

The online version of this article includes the following figure supplement(s) for figure 2:

**Figure supplement 1.** Yohimbine-induced changes in noradrenaline (NA) turnover and gene expression, and validation of opto- and chemogenetic NA release in the hippocampus.

**Figure supplement 2.** Effects of acute stress and noradrenergic stimulation on anxiety-like behavior in the open field test.

**Figure supplement 3.** Expression of *Dio2*, *Ppp1r3c*, *Ppp1r3g*, and *Sik1* is consistent across pharmacological, chemogenetic, and optogenetic manipulations of the noradrenergic system.

these transcriptional changes affected fewer genes than those observed after systemic noradrenergic activation by yohimbine administration.

Despite its specificity, chemogenetic LC activation does not provide the temporal control to restrict LC activation to shorter periods of time. Thus, we repeated the experiment using optogenetic LC activation (*Figure 2i*). To demonstrate that optogenetic LC activation directly triggers NA release in the hippocampus, we combined it with fiber photometry recordings using NA sensors (*Figure 2—figure supplement 1f*). We found that short (10 s) activation with 5 Hz triggered an almost instantaneous, sharp increase in hippocampal NA release in both females and males (*Figure 2—figure supplement 1g–h*). To mimic stress-induced LC activity, LC neurons were unilaterally stimulated with 5 Hz in a 3 min off/on paradigm for 21 min, which has previously been shown to phenocopy stress-induced effects

on behavior in mice (*McCall et al., 2015*; *McCall et al., 2017*). Again, tissue was collected 45 min after the start of stimulation, and in a separate cohort also 90 min afterward, to study how LC-mediated transcriptomic responses evolve over time. Optogenetic LC activation led to a significant cFos increase only in the stimulated LC hemisphere of ChR2+ animals, and these changes were significant at both time points (*Figure 2j*). Stimulated ChR2+ animals also showed a significant increase in the MHPG/NA ratio 45 and 90 min after stimulation onset compared to controls (*Figure 2k*). Additionally, we found that tonic 5 Hz activation of the LC led to immediate pupil dilation in ChR2+, but not in ChR2- animals (*Figure 2l*), as previously described (*Privitera et al., 2020*). Unilateral 5 Hz stimulation also reduced exploratory rearing behaviors in ChR2+ animals in the open field test (*Figure 2—figure supplement 2d*). Similar to the effects of acute stress and chemogenetic LC activation, unilateral 5 Hz stimulation of the LC induced significant transcriptomic changes at the 45 min time point in the ipsilateral vHC of ChR2+ mice compared to controls (*Figure 2m*). Notably, most of these changes disappeared again 90 min after stimulation onset, indicating that the LC-NA system mainly induces an early wave of transcriptomic changes in the hippocampus, which are not maintained over longer periods of time.

Next, we leveraged the extensive transcriptional data presented thus far to test which genes were consistently responsive to the various manipulations of the LC-NA system across experiments. First, we compared gene expression changes induced by yohimbine, chemogenetic, and optogenetic LC activation in the vHC. This allowed us to identify a small set of genes that are very consistently regulated across all modes of LC activation (*Figure 2n*). Second, we ranked genes across all the transcriptomic experiments according to their responsiveness to NA manipulations (based on p-value). This analysis includes acute stress exposure with pharmacological inhibition of NA receptors, as well as yohimbine treatment, chemogenetic and optogenetic LC activation. We then calculated the cumulative rank for each gene across all experiments to find genes with the most consistent response to NA manipulations (*Supplementary file 1*). This analysis reproduced most of the genes identified in *Figure 2n*, and additionally revealed more genes with particularly robust changes in response to LC-NA manipulations across experiments (*Figure 2o*, *Figure 2—figure supplement 3*). The top 10 genes were *Dio2*, *Ppp1r3c*, *Ppp1r3g*, *Sik1*, *Tsc22d3*, *Ppp1r3d*, *F3*, *Sertad1*, *Nr4a1*, and *Timp3*. For visualization, the top 4 of these genes are shown across all LC-NA manipulations in *Figure 2—figure supplement 3*. Going forward, we use these 10 transcripts as a bona fide list of LC-NA-responsive genes.

Our optogenetic data have demonstrated that LC neurons engage transcriptomic responses when firing at 5 Hz. Recent work has suggested that the effects of LC stimulation on brain processes, from behavior to brain network activity, depend on the firing pattern and frequency of the LC (*Harley and Yuan, 2021*; *Ghosh et al., 2021*; *Grimm et al., 2022*). To investigate if the molecular responses would differ between these stimulation conditions, we optogenetically activated the LC using two tonic paradigms (3 and 5 Hz), and one phasic paradigm (15 Hz, see schematic in *Figure 3a*). Stimulation was again conducted unilaterally in a 3 min off/on paradigm for 21 min for each of the stimulation groups, and tissue was collected for RNA sequencing 45 min after stimulation onset (*Figure 3a*). To increase statistical power in the face of higher variability due to the relatively small sample sizes (n=5–6 mice per group), we restricted our analysis to the 10 most LC-NA-responsive genes identified earlier. Surprisingly, we found that expression of these genes was similarly affected by tonic 3 Hz, 5 Hz and phasic 15 Hz LC stimulation (*Figure 3b*). These findings suggest that in contrast to circuit-wide changes, the molecular consequences to NA release are independent of the firing intensity and activity pattern of the LC neurons. While these transcriptomic changes seem to depend on LC-NA activity, our approach so far was not able to resolve whether NA mediates gene expression via direct effects within the hippocampus. Specifically, due to the widespread projections of the LC, it is possible that activation of other brain regions or other neurotransmitter systems might have led to indirect regulation of gene expression in the hippocampus. Thus, we selectively targeted only a subpopulation of LC neurons projecting to the hippocampus (LC$_{HC}$) using a unilateral, retrograde optogenetic approach (*Figure 3c*). Retrograde virus expression was restricted to dorsomedially located LC neurons ipsilateral to the injection site (*Figure 3d*), as previously described (*Robertson et al., 2013*). To confirm successful LC$_{HC}$ stimulation, we directly assessed NA turnover in the cortex and dHC. Indeed, 5 Hz stimulation of LC$_{HC}$ neurons led to an increased MHPG/NA ratio 45 min after stimulation onset in the ipsilateral dHC but not in the cortex (*Figure 3e*). In addition, 5 Hz stimulation of LC$_{HC}$ neurons did not impact pupil size, emphasizing the modular organization of the LC (*Figure 3f*). Within the vHC of the same animals we

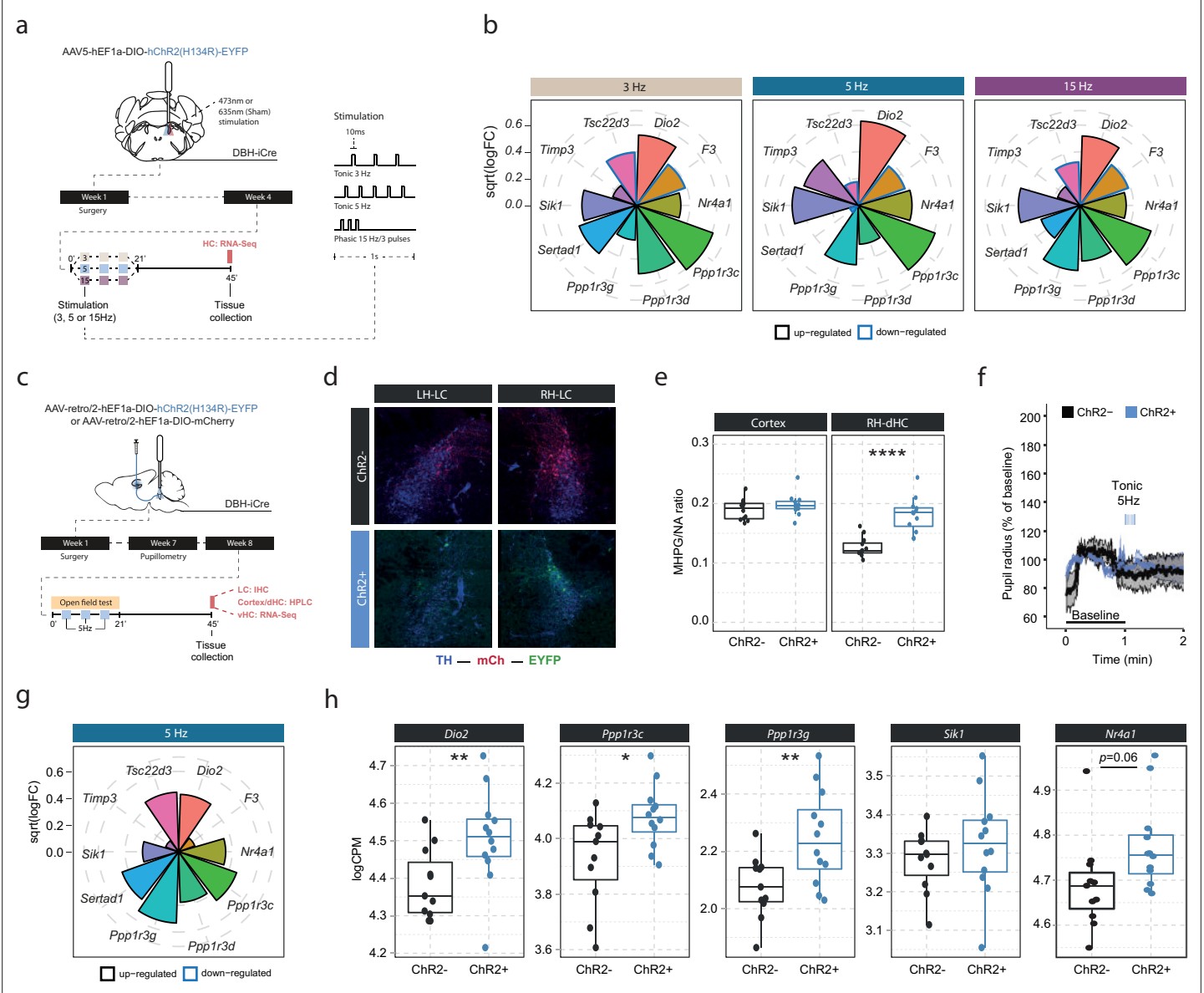

**Figure 3.** Locus coeruleus-noradrenaline (LC-NA)-mediated molecular responses in the hippocampus are independent of LC firing pattern and frequency and are directly stimulated via hippocampus-projecting LC neurons. (**a**) Experimental design for assessing molecular changes in the hippocampus induced by optogenetic LC activation with tonic (3 and 5 Hz) and phasic (15 Hz) firing patterns. (**b**) Radial plots showing expression changes (based on the logFC) of the most LC-NA-responsive genes after optogenetic LC activation in ChR2+ animals compared to controls (sham n = 6, 3 Hz n = 6, 5 Hz n = 7, 15 Hz n = 6). Black borders indicate that the gene is upregulated, blue border downregulated. (**c**) Experimental design for assessing molecular changes in the hippocampus induced by retrograde optogenetic 5 Hz activation of hippocampus-projecting LC neurons (LC$_{HC}$). (**d**) Representative images of retrograde mCherry (mCh, red) and ChR2-EYFP (EYFP, green) expression in tyrosine hydroxylase (TH, blue) positive LC neurons across hemispheres. (**e**) Cortical and right dorsal hippocampal (RH-dHC) NA turnover as measured by ultra-high performance liquid chromatography 45 min after 5 Hz optogenetic activation of LC$_{HC}$ neurons in ChR2- and ChR2+ animals. 5 Hz stimulation of LC$_{HC}$ neurons increased dHC but not cortical NA turnover in ChR2+ animals (unpaired t-test; t(17.43)=–5.5997, p=2.911e-05). ChR2-, n=12; ChR2+, n=12. ****p<0.0001. (**f**) Average pupil size changes in response to 5 Hz optogenetic activation of LC$_{HC}$-projecting neurons in ChR2- and ChR2+ animals. (**g**) Radial plots showing expression changes (based on the logFC) of the top 10 LC-NA-responsive genes in response to optogenetic LC$_{HC}$ activation with tonic 5 Hz stimulation in ChR2+ animals compared to ChR2- 45 min after stimulation onset (ChR2- n = 12, ChR2+ n = 12). (**h**) Selective boxplots of NA-responsive genes *Dio2*, *Ppp1r3c*, *Ppp1r3g*, *Sik1*, and *Nr4a1* in response to 5 Hz optogenetic activation of LC$_{HC}$-projecting neurons in ChR2- and ChR2+ animals 45 min after stimulation onset (ChR2- n = 12, ChR2+ n = 12). 5 Hz optogenetic activation of LC$_{HC}$-projecting neurons increased hippocampal expression of *Dio2*, *Ppp1r3c*, *Ppp1r3g*, and *Nr4a1*. *p<0.05, **p<0.01, ***p<0.001, ****p<0.0001.

then assessed the transcriptional impact of targeted LC$_{HC}$ 5 Hz activation on the top 10 NA-sensitive genes in the vHC at the 45 min time point. Indeed, activation of hippocampus-projecting LC neurons affected most target genes, including *Dio2, Ppp1r3c,* and *Ppp1r3g* (*Figure 3g–h*).

To understand in more detail how these NA-sensitive genes are affected by different stressors, we investigated whether the expression of *Dio2, Ppp1r3c, Ppp1r3g, Sik1,* and *Nr4a1* is specific to acute swim stress exposure or independent of the stress context. Therefore, we assessed their expression in a dataset comparing the effect of various stressors on the hippocampal transcriptome (*Floriou-Servou et al., 2018*). We found that these genes are not only upregulated by swim stress, but also by novelty stress, restraint, and footshock stress (*Figure 4b*). This suggests that expression of these genes is robustly induced by a wide range of stressors.

We then interrogated a recently published stress resource, which tracks stress-induced transcriptional changes over time in the hippocampus (*von Ziegler et al., 2022*). Across 4 hours following acute swim stress exposure, we found two distinctive expression patterns among these genes. While *Sik1* and *Nr4a1* show the characteristics of immediate early genes with a sharp rise and fall in expression within 90 min of stress onset, upregulation of *Dio2, Ppp1r3c,* and *Ppp1r3g* is maintained for at least 2–4 hr following stress exposure (*Figure 4d*), suggesting that mechanisms are in place to prolong expression beyond the initial rise in NA. Re-analysis of a hippocampal single-nucleus RNA sequencing dataset after a swim stress challenge (*von Ziegler et al., 2022*) revealed that stress-induced upregulation of *Dio2, Ppp1r3c,* and *Ppp1r3g* seems predominantly restricted to astrocytes, while *Sik1* and *Nr4a1* show a broader expression among glial, neuronal, and vascular cells.

Finally, we determined if these transcripts are also actively translated in astrocytes after stress exposure. We re-analyzed a dataset that used translating ribosome affinity purification followed by RNA sequencing (TRAPseq) in astrocytes of the somatosensory cortex, and assessed changes after a similar acute swim stress paradigm as described here (*Murphy-Royal et al., 2020*). We found that *Dio2, Ppp1r3c, Ppp1r3g,* and *Nr4a1* are significantly upregulated after stress exposure. Altogether, these results highlight that the NA-dependent gene expression changes that occur in response to stress exposures are most prominent in astrocytes.

## Discussion

### Dissecting stress with transcriptomics

The widespread molecular changes induced in the brain by an acute stress exposure (*Stankiewicz et al., 2015*; *von Ziegler et al., 2022*; *Floriou-Servou et al., 2018*; *Mifsud et al., 2021*) are part of a healthy stress response, and their dysregulation is often a hallmark of neuropsychiatric disorders (*Girgenti et al., 2021*; *Rubin et al., 2014*). To date, the contribution of corticosteroid signaling to stress-induced transcriptional changes has been well characterized (*Mifsud et al., 2021*; *Gray et al., 2014*; *Meijer et al., 2023*), yet the contribution of other stress mediators remains unexplored. Here, we extensively characterize the contribution of noradrenergic signaling to the transcriptomic response in the hippocampus during stress. By combining transcriptomics with circuit-specific manipulation of the LC-NA system, our unbiased approach reveals a small, but highly reproducible set of genes that are regulated directly by NA release from the LC. This gene set identifies astrocytes as a key target for NA-induced transcriptional changes.

### Complex interactions between stress mediators

Our results indicate that the transcriptomic response to a natural stressor is more complex than the gene expression changes induced solely by NA. This is well in line with the notion that multiple stress mediators contribute to the molecular response, and that these systems can also interact with each other. The response to LC-NA activation we observe in our experiments is short in duration. Following temporally precise optogenetic LC activation, gene expression changes did resolve within 90 min. This is noticeably different from an actual stress response, where gene expression evolves over a 4 hr period (*von Ziegler et al., 2022*). Specifically, LC-NA-regulated genes like *Ppp1r3c, Ppp1r3g,* and *Dio2* were elevated for several hours after stress exposure (*Figure 4d*). This suggests that other stress-induced signals can also regulate these genes more slowly or with a greater delay. Indeed, a recent study reported that activation of the glucocorticoid receptor by dexamethasone can induce strong transcriptomic changes 4 hr after injection across multiple brain regions (*Gerstner et al.,*

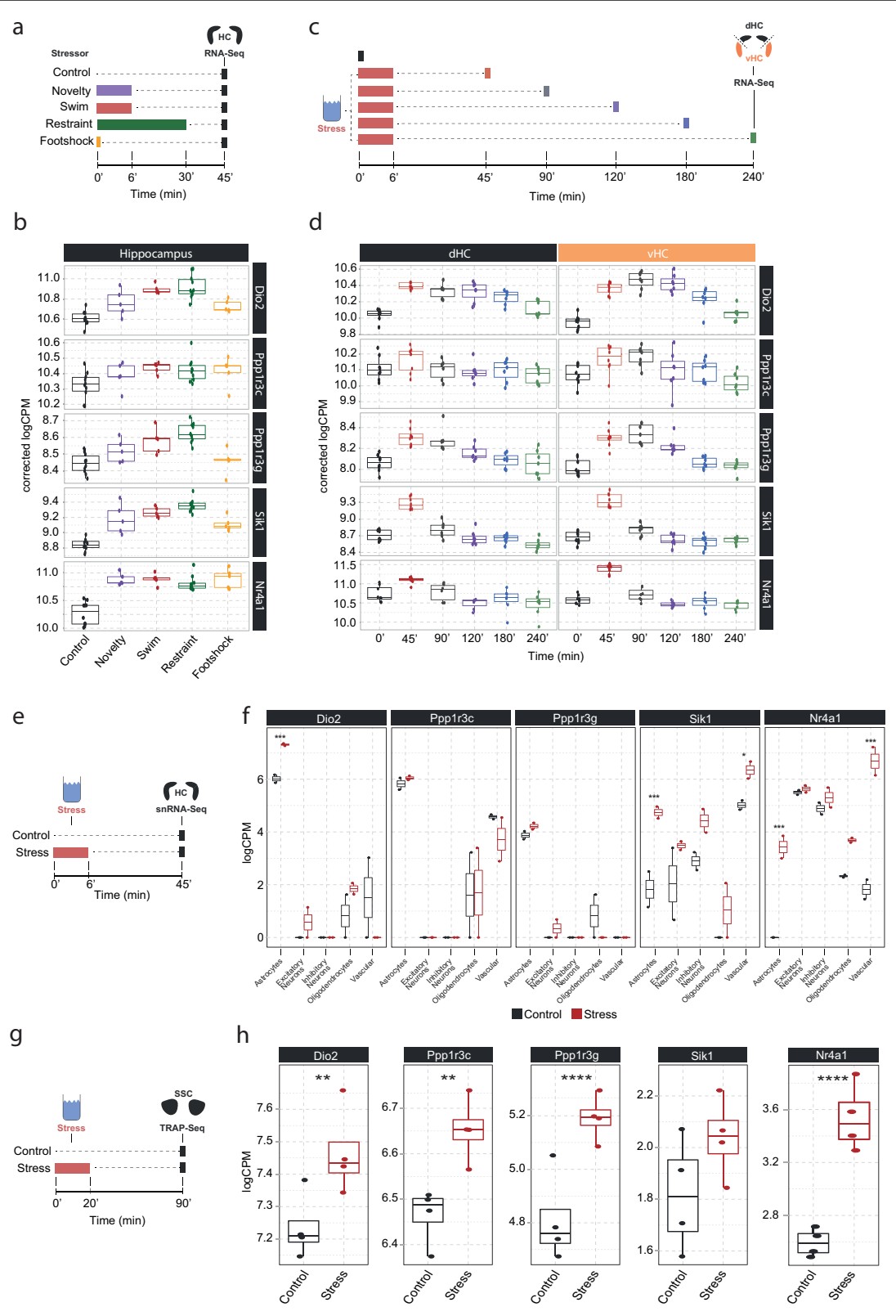

**Figure 4.** Screening of publicly available datasets shows that the noradrenaline-regulated genes *Dio2, Ppp1r3c, Ppp1r3g, Sik1,* and *Nr4a1* are induced by various stressors predominantly in astrocytes. (**a**) Experimental design for assessing transcriptomic changes in the hippocampus induced by different stressors as performed by **Floriou-Servou et al., 2018**. These stressors included a 10 min exposure to the open field test (novelty), a 6 min cold swim stress (swim), a 30 min immobilization stress (restraint), and exposure to a 1 mA footshock (footshock). (**b**) Selective boxplots of top NA-responsive

*Figure 4 continued on next page*

*Figure 4 continued*

genes *Dio2*, *Ppp1r3c*, *Ppp1r3g*, *Sik1*, and *Nr4a1* in response to different stressors. Control n = 10, novelty n = 5, swim n = 5, restraint n = 10, footshock n = 5. (**c**) Experimental design for assessing transcriptomic changes in the dorsal and ventral hippocampus across 4 hr following acute swim stress exposure as performed by von Ziegler et al. (**d**) Selective boxplots showing expression changes of top NA-responsive genes *Dio2*, *Ppp1r3c*, *Ppp1r3g*, *Sik1*, and *Nr4a1* across 4 hr following acute swim stress exposure. Control n = 8, 45 min n = 8, 90 min n = 7, 120 min n = 7, 180 min n = 7, 240 min n = 7. (**e**) Experimental design for assessing single-cell transcriptomic changes in the hippocampus 45 min following acute swim stress exposure by single-nucleus RNA sequencing as performed by *von Ziegler et al., 2022*. (**f**) Selective boxplots showing expression changes of top NA-responsive genes *Dio2*, *Ppp1r3c*, *Ppp1r3g*, *Sik1*, and *Nr4a1* across cell types of the hippocampus 45 min following acute swim stress exposure. Control n = 2, stress n = 2. (**g**) Experimental design for assessing actively translated RNA in the somatosensory cortex 90 min following a 20 min acute swim stress exposure by TRAP sequencing as performed by *Murphy-Royal et al., 2020*. (**h**) Selective boxplots of top NA-responsive genes *Dio2*, *Ppp1r3c*, *Ppp1r3g*, *Sik1*, and *Nr4a1* in the somatosensory cortex 90 min following a 20 min acute swim stress exposure. Acute stress increases the binding of *Dio2*, *Ppp1r3c*, *Ppp1r3g*, and *Nr4a1* mRNA to the ribosome. Control n = 4, stress n = 4. *p<0.05, **p<0.01, ***p<0.001, ****p<0.0001.

The online version of this article includes the following figure supplement(s) for figure 4:

**Figure supplement 1.** Sex-dependent expression of Ctla2b in the ventral hippocampus of female and male mice.

**Figure supplement 2.** Neuronal immediate early genes in the hippocampus are not regulated by noradrenaline signaling.

*2022*). Analyzing their data we found that *Dio2*, *Ppp1r3c*, and *Ppp1r3g* were all upregulated 4 hr after dexamethasone injection. This supports the concept that NA can act as a rapid molecular regulator, whereas glucocorticoid signaling can extend these stress-induced changes over longer time periods (*Hermans et al., 2014*).

In contrast to the small set of genes triggered by isolated LC-NA activation, blocking β-receptors prevents the induction of a large fraction of genes normally activated by natural stressors. This suggests that even if NA release alone is not sufficient to activate large numbers of genes, it is required to enable or enhance gene expression triggered via other mechanisms. A powerful regulator of transcription is neuronal activity linked to enhanced glutamate release (*Fernandez-Albert et al., 2019*; *Tyssowski et al., 2018*). The notion that NA release could enhance glutamate-dependent transcriptional cascades is in line with physiological evidence that NA can increase the excitability of neurons (*Bouret and Sara, 2002*), and with the 'glutamate amplifies noradrenergic effects' model, which posits that NA can amplify local glutamate release to create hotspots of activity (*Mather et al., 2016*).

Finally, our observation that systemic administration of the α2-adrenergic receptor antagonist yohimbine very closely recapitulates the transcriptional response to stress stands in contrast to the much more selective transcriptional changes observed after chemogenetic or optogenetic LC-NA activation. This difference could be due to various factors. First, it remains unclear how strong the LC gets activated by yohimbine versus hM3Dq-DREADDs. However, given the potent LC activation observed after DREADD activation, it seems unlikely that yohimbine would lead to a more pronounced LC activation, thus explaining the stronger transcriptional effects. Second, contrary to LC-specific DREADD activation, systemic yohimbine injection will also antagonize postsynaptic α2-adrenergic receptors throughout the brain (and periphery). More research is needed to determine whether this could have a more widespread impact on the hippocampus (and other brain regions) than isolated LC-NA activation, further enhancing excitability by preventing α2-mediated inhibition of cAMP production. Finally, systemic yohimbine administration and noradrenergic activity have been shown to induce corticosterone release into the blood (*Johnston et al., 1988*; *Leibowitz et al., 1988*; *Fink, 2016*). Thus, yohimbine injection could have broader transcriptional consequences, including corticosteroid-mediated effects on gene expression.

## Transcriptomic fingerprinting of NA effects using LC circuit manipulation

While systemic pharmacological treatments have been a common approach in studying the effects of different stress mediators and their receptors, they lack specificity and do not provide causal evidence that the release of a given stress mediator is sufficient to trigger molecular changes. By directly combining selective chemogenetic activation of the LC with transcriptomic analyses in the hippocampus, we were able to identify a subset of stress-responsive genes that depend on β-adrenergic signaling, and which can be triggered by NA alone in the absence of a physiological stress response. Using optogenetics we were able to validate these findings and further demonstrate that the strongest NA-mediated changes are similarly affected by tonic (3 and 5 Hz) and phasic (15 Hz) LC stimulation.

Interestingly, this is in contrast with our previous findings that these stimulation patterns differentially affect brain network connectivity (*Grimm et al., 2022*). This suggests that engagement of a transcriptomic response via β-adrenergic receptors seems common across these LC activity patterns, while changes on a network level might rely more on α1-mediated effects (*Zerbi et al., 2019*).

We found that direct activation of hippocampus-projecting LC neurons (LC$_{HC}$) was sufficient to increase the expression of the top NA-sensitive target genes, suggesting that local NA release in the hippocampus is directly contributing to these changes during stress. While we did not extensively characterize LC$_{HC}$ neurons, our data further show that in contrast to whole LC activation with 5 Hz, LC$_{HC}$ neurons do not seem to project to the cortex, nor do they affect pupil size.

## A uniform molecular response to stress and NA release in females and males

Our results, based on a meta-analysis of large sets of transcriptomic analyses across females and males, suggest that stress triggers a transcriptomic response in the hippocampus that is independent of sex. Extensive analyses of male and female hippocampi in response to stress and propranolol also suggest that stress- and propranolol-sensitive genes show similar response profiles in both sexes. However, due to the multivariate design and our genome-wide approach, subtle changes might not have survived multiple testing correction, particularly given that our study was not designed to track hormonal fluctuations across the estrous cycle, which is known to interact with NA signaling (*Bangasser et al., 2019*). An example is Ctla2b, which has previously been shown to be dependent on β-adrenergic signaling and to be selectively increased in females after stress exposure (*Roszkowski et al., 2016*). Indeed, targeted interrogation of our dataset confirms that Ctla2b is increased by stress only in females but not males, yet this effect fails to pass multiple testing correction (*Figure 4—figure supplement 1*). Ctla2b is also not included in the meta-analysis, as it did not pass the filtering step in our analysis pipeline due to very low baseline expression levels (see Methods). We provide a full list of genes that do not pass multiple testing correction, but still show nominally significant differences in gene expression (*Supplementary file 2*). Although these changes must be treated with caution, our data allow targeted hypothesis testing of individual genes to generate leads for follow-up work.

## Genes regulated by the LC-NA system

Across experiments, our transcriptomic screening revealed a conserved set of genes (*Figure 2o*, *Supplementary file 1*) that are selectively regulated by NA from LC projections in the hippocampus following acute stress exposure. Cross-referencing our data with publicly available single-cell databases suggests that - among the top 10 LC-NA sensitive genes - most are enriched in astrocytes (*Endo et al., 2022*; *von Ziegler et al., 2022*). Interestingly, immediate early genes commonly upregulated during stress and associated with neuronal activation like *Fos*, *Egr1*, *Arc*, *Dusp1*, and *Npas4* (*Smith et al., 1992*; *Benito and Barco, 2015*; *Fernandez-Albert et al., 2019*; *Tyssowski et al., 2018*) are not upregulated by LC stimulation (*Figure 4—figure supplement 2*). Taken together, our findings further add to accumulating evidence highlighting astrocytes as a direct and major cellular target of the LC-NA system (*Zenger et al., 2017*; *O'Donnell et al., 2015*; *Dienel, 2017*).

Within the hippocampus, astrocytic pathways are emerging as important players for learning and memory processes (*Gibbs et al., 2008*; *Bohmbach et al., 2022*). In fact, it is well known that NA enhances memory consolidation (*Schwabe et al., 2022*; *McGaugh and Roozendaal, 2002*), and recent work suggests that these effects are mediated by astrocytic β-adrenergic receptors (*Gao et al., 2016*; *Iqbal et al., 2023*). Our transcriptomic screens revealed *Dio2* as the most prominent target influenced by LC activity. *Dio2* is selectively expressed in astrocytes and encodes for the intracellular type II iodothyronine deiodinase, which converts thyroxine (T4) to the bioactive thyroid hormone 3,3',5-triiodothyronine (T3) and therefore regulates the local availability of T3 in the brain (*Bianco et al., 2019*). Enzymatic activity of DIO2 has further been shown to be increased by prolonged noradrenergic transmission through desipramine treatment in LC projection areas (*Campos-Barros et al., 1994*). This suggests that the LC-NA system and its widespread projections could act as a major regulator of brain-derived T3. Notably, T3 signaling plays a role in hippocampal memory formation (*Rivas and Naranjo, 2007*; *Sui et al., 2006*), raising the possibility that NA-induced *Dio2* activity in astrocytes might mediate some of these effects. Our molecular screening approach also revealed that three subunits of the astrocytic protein phosphatase 1 (*Ppp1r3d*, *Ppp1r3g*, and *Ppp1r3c*) respond

strongly to LC-NA activity. All three subunits enhance protein phosphatase 1-mediated glycogen synthesis. Especially *Ppp1r3c* expression has been found to be a master regulator of astrocytic glycogen synthesis and has previously been linked to NA activity (*Allaman et al., 2000*; *Petit et al., 2021*). Another important mechanism might include regulating sodium homeostasis via the widely expressed salt-inducible kinase 1 (*Sik1*). SIK1 has been shown to respond to neuronal activity and regulate Na+/K+-ATPase activity (*Jaitovich and Bertorello, 2010*; *Huang et al., 2012*; *Feldman et al., 2000*; *Bertorello and Zhu, 2009*). It was also found to detect low glucose availability and initiate gluconeogenesis in liver cells (*Wang et al., 2020*), a process which could also be important for noradrenergic activity in the brain. Our findings support the idea of the LC-NA system as a major regulator of brain-wide energy metabolism, stimulating astrocytic glycogen breakdown and consequently increasing energy supply to target areas (*Coggan et al., 2018*; *Dienel, 2017*).

Another interesting molecular target is the nuclear receptor subfamily 4 group A member 1 (Nr4a1), a widely expressed transcription factor that could trigger broader downstream changes. Within astrocytes, Nr4a1 activity was found to reduce oxidative stress and inflammation (*Dan et al., 2015*; *Popichak et al., 2018*) and might further regulate blood-brain barrier integrity (*Pan et al., 2021*; *Paillasse and de Medina, 2015*). Our re-analysis of published data showed that *Dio2*, *Ppp1r3c*, *Ppp1r3g*, and *Nr4a1* are actively translated in somatosensory cortical astrocytes following acute stress exposure. It remains to be tested whether protein levels, transcription factor activity, or enzymatic activity of these genes are also altered in the hippocampus, and what this ultimately means mechanistically for stress-related NA signaling.

## Summary

Overall, we provide the first genome-wide characterization of the molecular impact of NA release in vivo in the brain. The set of genes that are sensitive to NA release from the LC point to astrocytes as key molecular targets of NA during stress, and suggest that astrocytic processes involving glycogen and thyroid hormone metabolism could be key to the neuromodulatory effects of NA in the hippocampus.

## Methods

### Animals

All experiments were conducted in accordance with the Swiss federal guidelines for the use of animals in research and under licenses ZH161/17, ZH106/20 and ZH067/2022 approved by the Zurich Cantonal Veterinary Office. For experiments with wild-type animals, 2- to 3-month-old C57Bl/6J mice were obtained directly from Janvier (France). For experiments involving chemo- and optogenetic LC manipulations, heterozygous C57BL/6-Tg(Dbh-icre)1Gsc mice were kept in breeding trios with wild-type C57BL/6J mice at the ETH Zurich animal facility (EPIC). All mice were housed in groups of 2–5 per cage in a temperature- and humidity-controlled facility on a 12 hr reversed light-dark cycle (lights off: 9:15 am; lights on: 9:15 pm), with food and water ad libitum, and used for experiments at the age of 2–4 months. All experiments were conducted during the animals' active (dark) phase. For all experiments, mice were single-housed 24 hr before exposure to stress or LC activation, which reduces corticosterone levels in both sexes, and avoids confounding gene expression effects from social stressors (*Roszkowski et al., 2016*; *Bohacek et al., 2015*).

### Stereotactic surgeries

For experiments involving hippocampal infusions, female C57BL/6-Tg(Dbh-icre)1Gsc mice at the age of 2–3 months were subjected to stereotactic surgery. The mice were anesthetized with 4% isoflurane and then placed in a stereotaxic frame with continuous anesthesia of 2% isoflurane. For analgesia, animals received a subcutaneous injection of 5 mg/kg meloxicam and buprenorphine (0.1 mg/kg), as well as application of the local analgesics lidocaine (2 mg/kg) and bupivacaine (2 mg/kg) before and after surgery. After the skull was exposed, bregma (defined as the intersection of the coronal and sagittal suture) was located and the skull placement corrected for tilt and scaling. Bilateral holes were drilled above the hippocampus at –1.8 mm AP and ±1.5 mm ML from bregma, followed by the implantation of a bilateral guide cannula (62036, RWD Life Science, China) into the hippocampus (coordinates from bregma: –1.8 mm AP, ±1.5 mm ML, –1.5 mm DV).

For chemo- and optogenetic experiments male C57BL/6-Tg(Dbh-icre)1Gsc mice at the age of 2–3 months were subjected to stereotactic surgery. The mice were anesthetized with 4% isoflurane and then placed in a stereotaxic frame with continuous anesthesia of 2% isoflurane. For analgesia, animals received a subcutaneous injection of 5 mg/kg meloxicam and a local anesthetic (Emla cream; 5% lidocaine, 5% prilocaine) before and after surgery. After the skull was exposed, bregma was located and the skull placement corrected for tilt and scaling. Bilateral (chemogenetics) or unilateral (right hemisphere, optogenetics) small holes were drilled above the LC at –5.4 mm AP and 0.9 mm ML from bregma. A pneumatic injector (Narishige, IM-11–2) and calibrated microcapillaries (Sigma-Aldrich, P0549) were then used to inject 1 µl of virus to the LC (coordinates from bregma: –5.4 mm AP, ±1.0 mm ML, –3.8 mm DV). All viral vectors were obtained from the Viral Vector Facility (VVF) of the Neuroscience Center Zurich. For chemogenetic experiments, either ssAAV-5/2-hSyn1-dlox-hM3D(Gq)_mCherry(rev)-dlox-WPRE-hGHp(A) (hM3Dq+) or ssAAV-5/2-hSyn1-dlox-mCherry(rev)-dlox-WPRE-hGHp(A) (hM3Dq-) were injected bilaterally for all experiments except fiber photometry recordings where animals were injected unilaterally.

For optogenetic experiments, ssAAV-5/2-hEF1α-dlox-hChR2(H134R)_EYFP(rev)-dlox-WPRE-hGHp(A) (ChR2+) or ssAAV-5/2-hEF1a-dloxEGFP(rev)-dlox-WPRE-bGHp(A) (ChR2-) were delivered unilaterally to the right hemisphere locus coeruleus. For retrograde activation of hippocampus-projecting LC neurons, animals received one injection of either ssAAV-retro/2-hEF1a-dlox-hChR2(H134R)_EYFP(rev)-dlox-WPRE-hGHp(A) (ChR2+) or ssAAV-retro/2-hEF1a-dlox-mCherry(rev)-dlox-WPRE-hGHp(A) (ChR2-) to the ipsilateral dHC (coordinates from bregma: –2.10 mm AP, 1.5 mm ML; –1.8 mm DV) and vHC (coordinates from bregma: –3.30 mm AP, 2.75 mm ML; –4.0 mm DV). Additionally, all optogenetic animals were unilaterally implanted with an optic fiber (200 µm diameter, 0.22 NA) above the LC (coordinates from bregma: –5.4 mm AP, 0.9 mm ML, –3.5 mm DV). For fiber photometry recordings, animals were injected with the optogenetic actuator ChrimsonR (ssAAV-5/2-hEF1α/hTLV1-dlox-ChrimsonR_tdTomato(rev)-dlox-WPRE-bGHp(A); 4.7×10E12 vg/ml). Additionally, animals for both optogenetic and chemogenetic fiber photometry recordings received a second injection of a genetically encoded NA sensor (ssAAV-9/2-hSyn1-GRAB(NE1m)-WPRE-hGHp(A); 5.5×10E12 vg/ml, ssAAV-9/2-hSyn1- chI-nLightG-WPRE-bGHp(A); 5.5×10E12, or ssAAV-9/2-hSyn1-GRAB(NE2m)-WPRE-hGHp(A); 0.5×10E12 vg/ml) into the ipsilateral vHC (coordinates: AP –3.2 mm, ML –3.3 mm, DV –3.8 mm). To record from the vHC, an optic fiber was implanted 200 µm above the injection site (200 µm, NA = 0.37; Neurophotometrics, USA). The health of all animals was monitored over the course of 3 consecutive days post-surgery. Experiments on operated animals were conducted 4–8 weeks post-surgery to allow for recovery and sufficient virus expression. All viruses were obtained from the VVF of the University of Zurich and ETH Zurich.

## Drug injections/infusions

All drugs were freshly prepared immediately before experiments and dissolved in phosphate-buffered 0.9% saline (Gibco, pH 7.4). Drugs were administered intraperitoneally at their corresponding dosages: Yohimbine-hydrochloride (3 mg/kg, Merck, Germany), propranolol-hydrochloride (10 mg/kg, Merck, Germany), prazosin-hydrochloride (1 mg/kg, Merck, Germany), and clozapine (0.03 mg/kg, Merck, Germany).

For intra-hippocampal infusions of isoproterenol hydrochloride (Merck, Germany), animals were restrained and the guide cannula was inserted with an injector needle (62236, RWD Life Science, China) connected to an infusion pump (R462 Syringe Pump, RWD Life Science, China) via plastic tubing. Prior to attachment, the tubing was filled with sunflower seed oil (Merck, Germany) and vehicle (0.9% saline) or isoproterenol, separated by a small air bubble. Afterward, animals were allowed to freely roam their homecage for 2 min followed by bilateral intra-hippocampal infusions of vehicle drug or 1 µl of isoproterenol (3 µg/µl diluted in phosphate-buffered 0.9% saline) at 50 µl/min. Diffusion of vehicle and isoproterenol was allowed for another 2 min, before the animal was detached from the infusion setup and returned to its homecage.

## Fiber photometry

Fiber photometry recordings were performed as previously reported (*Grimm et al., 2022*). In short, the green fluorescence signal from the NA sensors GRAB$_{NE1m}$, GRAB$_{NE2m}$, or nLightG were recorded using a commercially available photometry system (Neurophotometrics, Model FP3002) controlled via

the open-source software Bonsai (v2.6.2). Two LEDs were used to deliver interleaved excitation light: a 470 nm LED for recording NA-dependent fluorescent signal ($F^{470}$) and a 415 nm LED for NA - independent control signals ($F^{415}$). Recording rate was set at 60 Hz for both LEDs allowing 30 Hz per channel. Excitation power at the fiber tip (200 µm, 0.39 numerical aperture; Doric Lenses) was set to 25–35 µW. Recordings during optogenetic LC stimulation were performed during light anesthesia (1.5% isoflurane) while chemogenetic LC activation was recorded in awake, well-handled animals.

### Forced swim test

For the forced cold swim stress, mice were placed for 6 min in a plastic beaker (20 cm diameter, 25 cm deep) filled with 18 ± 0.1°C water to 17 cm, in a room with dim red lighting. Immediately after stress exposure, mice returned to their assigned single-housing homecage.

### Open field test

Open field testing was performed in a square 45 cm (l)×45 cm (w)×40 cm (h) arena, and consisted of four black Plexiglas walls and a white PVC floor. Mice were tested under dim lighting (5 lux at the center of the arena). Mice were placed directly into the center of the open field and the tracking/recording was initiated 2 s after the mouse was detected. The test lasted 10 min for acute stress-exposed animals, 21 min for yohimbine and optogenetic stimulated animals, and 30 min for chemogenetic stimulated animals. Distance, time in center, supported and unsupported rearings were tracked by the software EthoVision XT14 (Noldus, Netherlands) and manual scoring. For pharmacological and chemogenetic experiments, animals received an i.p. injection of yohimbine (3 mg/kg) or clozapine (0.03 mg/kg) immediately before being placed into the arena. For optogenetic experiments, animals were attached to the optic fiber and directly placed into the arena.

### Optogenetic stimulation

Across optogenetic experiments the LC was stimulated with either 473 or 635 nm light at 10 mW power and 3, 5, or 15 Hz frequency (10 ms pulse width) alternating between 3 min off and on as previously described (*McCall et al., 2015*; *Grimm et al., 2022*).

### Pupillometry

Pupillometry was used to evaluate optogenetic LC stimulation as previously described (*Privitera et al., 2020*). At 3–4 weeks post-surgery, ChR2- and ChR2+ animals were anesthetized with 4% isoflurane and then placed in a stereotaxic frame with continuous anesthesia of 2% isoflurane. Recordings were performed on the right eye ipsilateral to the stimulated LC and consisted of an initial baseline recording of 60 s, followed by tonic LC stimulation (5 Hz at 10 mW for 10 s) and 1 min post-stimulation recording. Pupil videos were tracked with DeepLabCut and analyzed with the Pupillometry App.

### Tissue collection

At the appropriate time point after initiation of stress (for immediate groups within maximum 1 min after offset of stress) or LC activation, mice were euthanized by cervical dislocation and decapitation. The brain was quickly dissected on a cold glass plate and isolated hippocampi were separated by a cut at a ratio of 1:2 to divide the dHC and vHC. For experiments that were analyzed with uHPLC additionally the whole cortex was also collected. Isolated tissues were then snap-frozen in liquid nitrogen and stored at –80°C until further processing. For immunohistochemistry, the hindbrain (containing the LC) was isolated with a single cut from a razor blade at the beginning of the cerebellum and directly transferred to a tube with 4% PFA solution. All tissue dissections were performed between 11 am and 5 pm during the dark phase of the 12 hr reversed light-dark cycle.

### Immunohistochemistry

LC containing hindbrain samples were fixed for 2 hr in ice-cold paraformaldehyde solution (4% PFA in PBS, pH 7.4). The tissue was then rinsed with PBS and stored in a sucrose solution (30% sucrose in PBS) at 4°C, overnight. The tissue was frozen in tissue mounting medium (Tissue-Tek O.C.T. Compound, Sakura Finetek Europe B.V., Netherlands), and sectioned coronally using a cryostat (Leica CM3050 S, Leica Biosystems Nussloch GmbH) into 40 µm thick sections. The sections were immediately transferred into ice-cold PBS. LC containing sections were stained in primary antibody solution with 0.05%

Triton X-100, and 4% normal goat serum in PBS at 4°C under continuous agitation over two nights. The primary antibodies used were: mouse anti-TH (22941, Immunostar, 1:1000), chicken anti-GFP (ab13970, Abcam, 1:1000), and rabbit anti-cFOS (226 003, Synaptic Systems, 1:5000). Afterward, the sections were washed three times in PBS, and transferred to secondary antibody solution containing 0.05% Triton X-100, and 4% normal goat serum in PBS. The secondary antibodies used were: goat anti-mouse Alexa 488 (ab150113, Abcam, 1:300), goat anti-mouse Cy3 (115-165-003, Jackson ImmunoResearch, 1:300), goat anti-chicken Alexa 488 (A-11039, Thermo Fisher Scientific, 1:300), goat anti-rabbit Alexa 488 (A-11008 Thermo Fisher Scientific, 1:500), goat anti-rabbit Alexa 546 (A-11035, Thermo Fisher Scientific, 1:300), and donkey anti-mouse Alexa 647 (A-31571, Thermo Fisher Scientific, 1:300). Nissl was stained by NeuroTrace 640/660 Nissl stain (N21483, Thermo Fisher Scientific). After three more PBS washes, the sections were mounted onto glass slides (Menzel-Glaser SUPERFROST PLUS, Thermo Scientific), air-dried, and coverslipped with Dako fluorescence mounting medium (Agilent Technologies). Microscopy images were acquired in a confocal laser-scanning microscope (CLSM 880, Carl Zeiss AG, Germany) with a 20× objective. Images were analyzed using Fiji and for cFos quantification TH+ and cFos+ cells were counted manually.

## Ultra-high performance liquid chromatography

To quantify noradrenergic (NA; MHPG) compounds from cortical and hippocampal tissue, a reversed-phase uHPLC system coupled with electrochemical detection (RP-uHPLC-ECD) was used (Alexys Neurotransmitter Analyzer, Antec Leyden, Zoeterwoude, Netherlands). In short, our previously validated RP-HPLC method with ion pairing chromatography was applied as described (*Van Dam et al., 2014*), albeit with minor modifications regarding the installed column (BEH C18 Waters column, 150 mm × 1 mm, 1.7 µm particle size) and pump preference (LC110S pump, 470–480 bar; flow rate of 62 µl/min), achieving the most optimal separation conditions in an RP-UHPLC setting. Brain samples were defrosted to 4°C and subsequently homogenized in 800–900 µl ice-cold sample buffer (50 mM citric acid, 50 mM phosphoric acid, 0.1 mM EDTA, 8 mM KCl, and 1.8 mM octane-1-sulfonic acid sodium salt, adjusted to pH = 3.6), using a Precellys Minilys Personal Tissue Homogenizer (Bertin Technologies, France) with CK14 1.4 mm ceramic beads (40–60 s approximately, full speed). To remove excess proteins, 450 µl homogenate was transferred onto a 10,000 Da Amicon Ultra 0.5 Centrifugal Filter (Millipore, Ireland) that had been pre-washed twice using 450 µl sample buffer (centrifugation: $14{,}000 \times g$, 20 min, 4°C). The Amicon filter loaded with the homogenate was then centrifuged ($14{,}000 \times g$, 20 min, 4°C). Finally, the filtrate was transferred into a polypropylene vial (0.3 ml, Machery-Nagel GmbH & Co. KG, Germany) and automatically injected into the uHPLC column by the Alexys AS110 Autosampler (5 µl sample loop). Levels of the monoamines and metabolites were calculated using Clarity software (DataApex Ltd., v86.12.0.77208, 2015, Prague, Czech Republic).

## RNA extraction

dHC and vHC samples were homogenized in 500 µl Trizol (Invitrogen 15596026) in a tissue lyser bead mill (QIAGEN, Germany) at 4°C for 2 min, and RNA was extracted according to the manufacturer's recommendations. This was followed by determining RNA purity and quantity with a UV/V spectrophotometer (NanoDrop 1000).

## Bulk RNA sequencing and data analysis

For experiments shown in *Figures 1c–e and 2*, bulk mRNA sequencing was performed at the Functional Genomics Center Zurich (FGCZ) core facility. Data shown in *Figure 1c–e* and *Figure 2c–d* belong to the same experiment and were split up for better visualization of effects after sample processing and RNA sequencing analysis was performed. RNA integrity was assessed with high sensitivity RNA screen tape on an Agilent Tape Station/Bioanalyzer, according to the manufacturer's protocol. The RIN values of all samples ranged from 8.4 to 10.0. For library preparation, the TruSeq stranded RNA kit (Illumina Inc) was used according to the manufacturer's protocol. For bulk sequencing library preparation, the TruSeq stranded RNA kit (Illumina Inc) was used according to the manufacturer's protocol. The mRNA was purified by polyA selection, chemically fragmented and transcribed into cDNA before adapter ligation. Single-end (100 nt) sequencing was performed with HiSeq 4000. Samples within experiments were each run on one or multiple lanes and demultiplexed. A sequencing depth of ~20M reads per sample was used. Bulk mRNA sequencing for experiments shown in *Figures 1g–h and 3* were

performed at Novogene UK. Total RNA samples were used for library preparation using NEB Next Ultra RNA Library Prep Kit for Illumina. Indices were included to multiplex multiple samples. Briefly, mRNA was purified from total RNA using poly-T oligo-attached magnetic beads. After fragmentation, the first strand cDNA was synthesized using random hexamer primers followed by the second strand cDNA synthesis. The library was ready after end repair, A-tailing, adapter ligation, and size selection. After amplification and purification, insert size of the library was validated on an Agilent 2100 and quantified using quantitative PCR. Libraries were then sequenced on Illumina NovaSeq 6000 S4 flow-cell with PE150 according to results from library quality control and expected data volume. Samples within experiments were each run on one or multiple lanes and demultiplexed. A sequencing depth of ~40M reads per sample was used.

For all experiments, adapters were trimmed using cutadapt (*Martin, 2011*) with a maximum error rate of 0.05 and a minimum length of 15. Kallisto (v0.44.0) (*Bray et al., 2016*) was used for pseudo alignment of reads on the transcriptome level using the genecode.vM17 assembly with 30 bootstrap samples and an estimated fragment length of 200±20 for single-end samples. For differential gene expression (DGE) analysis we aggregated reads of protein coding transcripts and used R (v. 4.0.3) with the package 'edgeR' (v 3.32.1) for analysis. A filter was used to remove genes with low expression prior to DGE analysis. EdgeR was then used to calculate the normalization factors (TMM method) and estimate the dispersion (by weighted likelihood empirical Bayes). For two group comparisons the genewise exact test was used, for more complex designs we used a generalized linear model (GLM) with empirical Bayes quasi-likelihood F-tests. Exact specifications for each tested model can be found under https://github.com/ETHZ-INS/Privitera-et.-al.-2023 (data deposited at https://doi.org/10.5281/zenodo.10730454). For multiple testing correction the Benjamini-Hochberg FDR method was used, and for interaction terms (e.g. stress:sex or stress:injection), FDR was calculated only on the genes that have a significant stress effect. For analyses of datasets originating from multiple experiments we further employed SVA correction to correct for processing specific effects (*Leek et al., 2012*). Surrogate variables independent of experimental groups were identified using the SVA package (v3.38.0) on data after DESeq2 (v1.30.1) variance stabilization (*Love et al., 2014*), and were then included as additive terms in the GLMs. Heatmaps were produced with the sechm (v1.5.1) package. To avoid rare extreme values from driving the scale, the color scale is linear for values within a 98% interval, and ordinal for values outside it. Unless otherwise specified, the rows were sorted using the features' angle on a two-dimensional projection of the plotted values, as default in sechm.

For the combined analysis of consistent effects across yohimbine injection, chemogenetic and optogenetic stimulation, we first combined all three datasets and modeled batch effects using SVA correction. We then designed a combined response variable that was set to control (homecage in the injection experiment, hM3Dq- in chemogenetic and ChR2- in optogenetic) or response (yohimbine in the injection experiment, hM3Dq+ in chemogenetic and ChR2+ in optogenetic). We then fit an additive GLM with the newly defined response variable and the surrogate variables from the SVA correction and tested it for the response variable coefficient.

For the cumulative rank analysis, statistical results were used from multiple analyses (stress group vs propranolol group in vHC of the first injection experiment; stress group vs propranolol group in dHC of the first injection experiment; stress:propranolol interaction in second injection experiment; effect of chemogenetic LC activation in vHC; effect of chemogenetic LC activation in dHC; effect of optogenetic LC activation after 45 min). Then, in each analysis the gene with the lowest p-value was set to rank 1, the one with the highest to rank N. These ranks were then summed up across all analyses to generate the cumulative rank.

### Reverse transcription quantitative real-time polymerase chain reaction

Reactions were conducted using SYBR green (Roche) on a CFX384 Touch Real-Time PCR Detection System (Bio-Rad) and normalized against Tubulin delta 1 (Tubd1). Cycling conditions were 5 min at 95°C, then 50 cycles with denaturation (10 s at 95°C), annealing (10 s at 60°C), and elongation (10 s at 72°C). Primers were designed using PrimerBlast (*Kozyreva et al., 2021*) and tested for quality and specificity by melt curve analysis, gel electrophoresis, and appropriate negative controls. Forward (FP) and reverse (RP) primer sequences were as follows:

Tubd1: FP: TCTCTTGCTAACTTGGTGGTCCTC / RP: GCTGGGTCTTTAAATCCCTCTACG
Apold1: FP: ACCTCAGGCTCTCCTTCCATCATC / RP: ACCCGAGACAAAGCACCAATGC

Dio2: FP: GCCTACAAACAGGTTAAACTGGGTG / RP: CCATCAGCGGTCTTCTCC
Sik1: FP: ACAGCTCACTTCAGCCCTTAT / RP: CTCGCTGATAGCTGTGTCCA
Ppp1r3c: FP: TGAGCTGCACCAGAATGATCC / RP:GGTGGTGAATGAGCCAAGCA

## Statistics

We used a block design for experiments. Animals and samples were split into multiple blocks, containing one replicate of each condition. Experimental and processing order within these blocks was randomized. Investigators were blinded during behavior and sample processing, but not during the analysis process. However, the same algorithmic analysis methods were used for all samples within each sequencing experiment. Analysis was performed in R or GraphPad Prism 9.2.0. For statistical analyses of behavior, pupillometry, immunohistochemistry, and uHPLC data, we used independent sample t-tests when comparing two independent groups. When comparing more than two groups, we used one-way ANOVAs if there was a single independent variable, or two-way ANOVAs for two-factorial designs (e.g. injection × group). Significant main effects and interactions were analyzed using Tukey's post hoc tests. For linear model analysis we used the function lm() from the 'stats' package in R and F-statistics for significance testing. No statistical method was used to predetermine sample size. No data were excluded from the analyses.

## Code availability

Code for all analyses (independent scripts) presented here is available on GitHub under https://github.com/ETHZ-INS/Privitera-et.-al.-2023; (copy archived at *Privitera et al., 2023*).

## Acknowledgements

The lab of JB was supported by the ETH Zurich, ETH Project Grant ETH-20 19-1, the Swiss National Science Foundation (grants 310030_172889/1 and 310030_204372), the Basel Research Centre for Child Health, the Swiss 3R Competence Center, Roche, the Hochschulmedizin Zürich Flagship project STRESS, the Forschungskredit of the University of Zurich, the Novartis Foundation for Medical-Biological Research, the Swiss Foundation for Excellence and Talent in Biomedical Research, the Vontobel Foundation, the Betty and David Koetser Foundation for Brain Research. The lab of PPDD and DVD was supported by the University of Antwerp, Research Foundation Flanders (FWO), Joint Programming Initiative Neurodegenerative Diseases (JPND) and ZonMW (HEROES 73305172), Alzheimer Nederland and Neurosearch Antwerp. We thank the staff of the EPIC for the excellent animal care and their service to our animal facility and Prof. Isabelle Mansuy for providing support and space. We thank Dr. Maria Wilhelm for photometry data analysis, Han-Yu Lin and Justine Leonardi for help with sample processing and Julia Bode for maintaining the animal colony.

## Additional information

### Funding

| Funder | Grant reference number | Author |
| --- | --- | --- |
| ETH Zurich | Base Funding | Johannes Bohacek |
| ETH Zurich | ETH Project Grant ETH-20 19-1 | Johannes Bohacek |
| Swiss National Science Foundation | grants 310030_172889/1 and 310030_204372 | Johannes Bohacek |
| Basel Research Centre for Child Health | Multi-Investigator Program | Johannes Bohacek |
| Swiss 3R Competence Center | OC-2019-003 | Johannes Bohacek |
| Hochschulmedizin Zürich | Flagship Project STRESS | Johannes Bohacek |

| Funder | Grant reference number | Author |
| --- | --- | --- |
| Forschungskredit of the University of Zurich | FK-15-035 | Johannes Bohacek |
| Novartis Foundation for medical-biological research | #16A016 | Johannes Bohacek |
| Vontobel-Stiftung | | Johannes Bohacek |
| Betty and David Koetser Foundation for Brain Research | | Johannes Bohacek |
| Research Foundation Flanders | FWO | Peter P De Deyn Debby Van Dam |
| Joint Programming Initiative Neurodegenerative Diseases | | Peter P De Deyn Debby Van Dam |
| Alzheimer Nederland | | Peter P De Deyn Debby Van Dam |
| ZonMw | HEROES 73305172 | Peter P De Deyn Debby Van Dam |
| Neurosearch Antwerp | | Peter P De Deyn Debby Van Dam |

The funders had no role in study design, data collection and interpretation, or the decision to submit the work for publication.

## Author contributions

Mattia Privitera, Conceptualization, Data curation, Formal analysis, Methodology, Writing – original draft, Writing – review and editing; Lukas M von Ziegler, Conceptualization, Software, Formal analysis, Methodology, Writing – original draft; Amalia Floriou-Servou, Conceptualization, Formal analysis, Investigation, Methodology; Sian N Duss, Formal analysis, Investigation, Visualization, Methodology, Writing – review and editing; Runzhong Zhang, Sebastian Leimbacher, Investigation; Rebecca Waag, Investigation, Methodology, Writing – review and editing; Oliver Sturman, Investigation, Visualization, Methodology; Fabienne K Roessler, Data curation, Visualization; Annelies Heylen, Debby Van Dam, Investigation, Methodology; Yannick Vermeiren, Resources, Formal analysis, Supervision, Funding acquisition, Investigation, Methodology, Writing – original draft; Peter P De Deyn, Resources, Supervision, Funding acquisition; Pierre-Luc Germain, Data curation, Software, Formal analysis, Supervision, Investigation, Visualization, Methodology, Writing – original draft, Writing – review and editing; Johannes Bohacek, Conceptualization, Resources, Supervision, Funding acquisition, Writing – original draft, Project administration, Writing – review and editing

## Author ORCIDs

Mattia Privitera ⬤ http://orcid.org/0000-0002-5660-5901
Amalia Floriou-Servou ⬤ http://orcid.org/0000-0002-5090-4900
Sian N Duss ⬤ http://orcid.org/0000-0003-0212-0324
Runzhong Zhang ⬤ http://orcid.org/0000-0001-6144-5546
Rebecca Waag ⬤ http://orcid.org/0000-0001-5103-2860
Oliver Sturman ⬤ http://orcid.org/0000-0001-6859-4800
Fabienne K Roessler ⬤ http://orcid.org/0000-0001-6594-1504
Annelies Heylen ⬤ http://orcid.org/0000-0002-2660-8984
Yannick Vermeiren ⬤ http://orcid.org/0000-0002-2091-5326
Debby Van Dam ⬤ http://orcid.org/0000-0003-4739-6076
Peter P De Deyn ⬤ http://orcid.org/0000-0002-2228-2964
Pierre-Luc Germain ⬤ http://orcid.org/0000-0003-3418-4218
Johannes Bohacek ⬤ http://orcid.org/0000-0002-8442-653X

## Ethics

All experiments were conducted in accordance with the Swiss federal guidelines for the use of animals in research and under licenses ZH161/17, ZH106/20 and ZH067/2022 approved by the Zurich Cantonal Veterinary Office.

Reviewer #2 (Public Review): https://doi.org/10.7554/eLife.88559.3.sa1
Author Response https://doi.org/10.7554/eLife.88559.3.sa2

## Additional files

### Supplementary files

• Supplementary file 1. List of reproducible gene expression changes across various manipulations of the locus coeruleus-noradrenaline (LC-NA) system. This table displays the logarithmic cumulative rank of genes across all experiments from *Figure 1* and *Figure 2* in terms of their responsiveness to manipulations of the LC-NA system. A lower cumulative rank indicates that a gene is among the more significant hits across all analyses (for full list of included analyses, see Methods). The top 10 genes from this list are shown in *Figure 2o*.

• Supplementary file 2. Sex-specific regulation of transcription in the ventral hippocampus after swim stress exposure. Differential expression analysis results for a sex:stress interaction performed across the meta-analysis of vHC datasets 45 min after stress initiation (total of 50 males and 20 females). The table merges the results of both the edgeR and the DESeq2 analyses. Detailed information is provided in the sheet 'ColumnDefinitions'.

• MDAR checklist

### Data availability

The sequencing data data generated in this study have been deposited in the Gene Expression Omnibus database under accession code GSE218315 for all injection experiments and GSE218313 for chemo and optogenetic experiments. Code for all analyses (independent scripts) presented here is available on GitHub under https://github.com/ETHZ-INS/Privitera-et.-al.-2023; copy archived at *Privitera et al., 2023*.

The following datasets were generated:

| Author(s) | Year | Dataset title | Dataset URL | Database and Identifier |
| --- | --- | --- | --- | --- |
| von Ziegler L, Privitera M, Germain P, Bohacek J | 2023 | Effects of pharmacological manipulations of the NA system on the transcriptional stress response in the mouse hippocampus | https://www.ncbi.nlm.nih.gov/geo/query/acc.cgi?acc=GSE218315 | NCBI Gene Expression Omnibus, GSE218315 |
| Privitera M, von Ziegler L, Duss S, Bohacek J | 2023 | Effects of optogenetic and chemogenetic LC activation on the hippocampal transcriptome | https://www.ncbi.nlm.nih.gov/geo/query/acc.cgi?acc=GSE218313 | NCBI Gene Expression Omnibus, GSE218313 |
| von Ziegler L, Privitera M, Germain P-L, Bohacek J | 2024 | ETHZ-INS/Privitera-et.-al.-2023: eLife | https://doi.org/10.5281/zenodo.10730453 | Zenodo, 10.5281/zenodo.10730453 |

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
