## [Editor Report · eLife assessment]

This **important** paper uses a multifaceted approach to implicate the locus coeruleus-noradrenaline system in the stress-induced transcriptional changes of dorsal and ventral hippocampus. It provides an inventory of dorsal and ventral hippocampal gene expression upregulated by activation of LC-NA system, which can be used as starting point for more functional studies related to the effects of stress-induced physiological and pathological changes. The results **convincingly** support the conclusions. This paper will be of interest to those interested in stress neurobiology, hippocampal, and/or noradrenaline function.

---

## [Referee Report · Reviewer #2 (Public Review)]

The present manuscript investigates the implication of locus coeruleus-noradrenaline system in the stress-induced transcriptional changes of dorsal and ventral hippocampus, combining pharmacological, chemogenetic, and optogenetic techniques. Authors have revealed that stress-induced release of noradrenaline from locus coeruleus plays a modulatory role in the expression of a large scale of genes in both ventral and dorsal hippocampus through activation of β-adrenoreceptors. Similar transcriptional responses were observed after optogenetic and chemogenetic stimulation of locus coeruleus. Among all the genes analysed, authors identified the most affected ones in response to locus coeruleus-noradrenaline stimulation as being Dio2, Ppp1r3c, Ppp1r3g, Sik1, and Nr4a1. By comparing their transcriptomic data with publicly available datasets, authors revealed that these genes were upregulated upon exposure to different stressors. Additionally, authors found that upregulation of Ppp1r3c, Ppp1r3g, and Dio2 genes following swim stress was sustained from 90 min up to 2-4 hours after stress and that it was predominantly restricted to hippocampal astrocytes, while Sik1 and Nr4a1 genes showed a broader cellular expression and a sharp rise and fall in expression, within 90 min of stress onset.

The paper is well written and provides a useful inventory of dorsal and ventral hippocampal gene expression upregulated by activation of LC-NA system, which can be used as starting point for more functional studies related to the effects of stress-induced physiological and pathological changes. Sex-differences were also explored which represents a strength of the study.

---

## [Author Response]

The following is the authors’ response to the original reviews.

**eLife assessment**
This manuscript provides novel and important findings regarding the impact of noradrenergic signaling from the locus coeruleus on hippocampal gene expression. The locus coeruleus is the sole source of noradrenaline to the hippocampus and many rapid molecular changes induced by stress are regulated by noradrenaline. This manuscript provides a rigorous investigation into hippocampal genes uniquely regulated by noradrenaline in the presence or absence of stress. Data were collected and analyses were performed using solid methodology, and the results mostly convincingly support the conclusion made with few weaknesses. The study would benefit from a more comprehensive analyses of sex differences.

Response: We thank the reviewers and the editors for the positive evaluation of our work and for the constructive feedback. To address some of the key criticisms, we have performed several new experiments and analyses. Importantly, we now provide a much more rigorous comparison of males and females, which strongly suggests that there are no major sex differences in the transcriptomic response to stress and noradrenaline in the hippocampus. We think that these - and other additions discussed below - significantly strengthen the manuscript. We provide detailed responses to all the reviewers comments. We have added numbers to the reviewers’ comments for easier referencing.

**Reviewer #1 (Public Review):**
Comment 1: Privitera et al., provide a comprehensive and rigorous assessment of how noradrenaline (NA) inputs from the locus coeruleus (LC) to the hippocampus regulate stress-induced acute changes in gene expression. They utilize RNA-sequencing with selective activation/inhibition of LC-NA activity using pharmacological, chemogenetic and optogenetic manipulations to identify a great number of reproducible sets of genes impacted by LC activation. It is noteworthy that this study compares transcriptomic changes in the hippocampus induced by stress alone, as compared with selective circuit activation/inhibition. This reveals a small set of genes that were found to be highly reproducible. Further, the publicly available data will be highly useful to the scientific community.

Response: We are very grateful for this positive evaluation.

Comment 2: A major strength of the study is the inclusion of both males and females. However, with this aspect of the study also lies the biggest weakness. While the experiments tested males and females, they were not powered for identifying sex differences. There are vast amounts of literature documenting the inherent sex differences, both under resting and stress-evoked conditions, in the LC-NA system and this is a major missed opportunity to better understand if there is an impact of these sex-specific differences at the genetic level in a major LC projection region. There are many instances whereby sex effects are apparent, but do not pass multiple testing correction due to low n's. The authors highlight one of them (Ctla2b) in supplemental figure 6. This gene is only upregulated by stress in females. It is appreciated that the manuscript provides an incredible amount of novel data, making the investigation of sex differences ambitious. Data are publicly available for others to conduct follow up work, and therefore it may be useful if a list of those genes that were different based on targeted interrogation of the dataset be provided with a clear statement that multiple testing corrections failed. This will aid further investigations that are powered to evaluate sex effects.

Response: The assessment of the reviewers and the editorial feedback encouraged us to look more thoroughly into potential sex differences, because we believe it would indeed be a major additional strength if our manuscript could make a firm statement on this important issue. To this end, we have expanded the manuscript in two major ways:

(1) To expand the analysis of sex effects also to the dorsal hippocampus, and to increase robustness of the data, we have performed RNA-seq in 32 additional samples of male and female mice exposed to stress (or control) and propranolol (or saline) injection. Figure 1fh and Supplementary Figure 1d-f have been updated to reflect this new addition, and the results are presented in a new section on Pages 3-4 (pasted below for ease of reviewing). In summary, the strongly support our initial observation that the effects of stress on gene expression, as well as the effects of propranolol on blocking stress-induced effects, are highly similar in both sexes.

(2) To further increase the power for detection of sex-effects, we have performed a small meta-analysis. For this, we combined several RNAseq datasets from the current manuscript and published datasets from our previous work (Floriou-Servou et al., 2018; von Ziegler et al., 2022), which also investigated transcriptomic sex-differences in the hippocampus 45 min after cold swim stress exposure in the same setup as used for the current manuscript. This approach increased our sample size to 51 males and 20 females. In summary, this well-powered approach shows no evidence for sex differences in the transcriptional response to stress, even when more lenient analyses were applied. These results are described in a new section on page 4, and summarized in Supplementary Figures 1f+g. This section is pasted below for ease of reviewing.

"While blocking β-adrenergic receptors was able to block stress-induced gene expression, we did not test whether propranolol might decrease gene expression already at baseline, independent of stress. Additionally, all tests had thus far been conducted in male mice, raising the question about potential sex differences in NA-mediated transcriptomic responses. To address these two issues, we repeated the experiment in both sexes and included a group that received a propranolol injection but was not exposed to stress (Fig. 1f). Combining the data from both experiments, we repeated the analysis for each region, to identify genes whose response to stress was inhibited by propranolol (Figure 1g). As in the previous experiment, we found that many of the stress-induced gene expression changes were blocked by propranolol injection in both dHC (Figure 1g, left panel) and vHC (Figure 1g, right panel). Importantly, propranolol did not change the expression level of these genes in the absence of stress. We then directly compared the genes sensitive to stress and propranolol treatment in both dHC and vHC. To this end, we plotted the union of genes showing a significant stress:propranolol interaction in either region in one heatmap across both dHC and vHC (Supplementary Figure 1d). This showed again that the stress-induced changes were very similar in dHC and vHC, and that propranolol similarly blocked many of them. Finally, we asked whether the response differs between males and females. Despite clear sex differences in gene expression at baseline (data not shown), we found no significant sex differences in response to stress or propranolol between male and female mice (FDR<0.05; Fig. 1g). To more directly visualize this, we compared females and males by plotting the log2-fold changes of the stress:propranolol interaction across all stress-induced genes that were blocked by propranolol. We find very similar regulation patterns in both sexes (Figure 1h). Although none of these sex differences are significant, some genes seem to show quantitative differences, so we plotted the expression patterns of the 5 genes showing the largest difference in interaction term as box-plots, which suggest that these spurious differences are likely due to noisy coefficient estimates (Supplementary Fig. 1e). To address concerns that our analysis of sex differences might not have been sufficiently powered, we performed a meta-analysis of the experiments shown here along with previously published datasets from our lab (Floriou-Servou et al. 2018; von Ziegler et al. 2022). In all these experiments, the vHC of male and female mice was profiled 45 min after exposure to an acute swim stress challenge. This resulted in a sample size of 51 males and 20 females. Despite this high number of independent samples, we could not identify any statistically significant interaction between sex and the stress response. To identify candidates that might not reach significance while discounting differences due to noise in fold-change estimates, we reproduced the same analysis using DESeq2 with Approximate Posterior Estimation for generalized linear model (apeglm) logFC shrinkage (A. Zhu, Ibrahim, and Love 2018). This analysis also did not reveal any sex differences in the stress response (Supplementary Fig. 1f). We then tailored the meta-analysis specifically to the set of stress-responsive genes that were blocked by propranolol, and also for these genes the response to stress was strikingly similar in both sexes (Supplementary Fig. 1g). Altogether, we conclude that there are no major sex differences in the rapid transcriptomic stress response in the hippocampus, and that blocking beta-receptors prevents a large set of stress-induced genes in both females and males."

To put these findings in context with existing literature, we agree with the reviewer that there are many studies that have reported sex differences in the LC-circuitry as summarized by Bangasser and colleagues (Bangasser et al., 2016, 2019). However, these studies primarily focus on the LC itself, suggesting that female rats have more LC neurons, denser LC-dendrites in the peri-LC region, and that LC neurons are more readily activated by stress in females because of heightened sensitivity to CRF-signaling. A recent study in mice reports, in contrast, that females have fewer TH-positive neurons in the LC, but they also find enhanced excitability of LC neurons in females (Mariscal et al., 2023). Similarly, one study has suggested molecular differences in the makeup of the LC (Mulvey et al., 2018). Our experiments, however, focus on the impact of NA release in a projection region (hippocampus). Further, we use a strong stress induction protocol (swim stress) and various potent modes of direct LC activation, so differences in "LC-excitability" are likely less relevant in this context. We added evidence showing that we trigger powerful NA release in both sexes (Supplementary Figure 2c-h; see response to Reviewer #2, Comment #3 for more details). In addition, we show that the intensity or pattern of LC stimulation does not appear to alter the molecular response (Figure 3a-b), and that various stressors (mild or intense) all trigger the same NA-dependent molecular changes (Figure 4a-b). Therefore, our results suggest that once NA is released (in the hippocampus), the molecular downstream effects on gene expression are very similar - independent of stimulation intensity, sex, or hippocampal subregion (dorsal/ventral). This does not mean that there are no sex differences for activation of LC, but rather that the transcriptional response to NA release in the hippocampus is robust across sexes, and that propranolol seems to block NA-dependent effects similarly in both sexes. This does not rule out quantitative differences between sexes that only emerge with targeted analyses of individual genes, or once fluctuations in ovarian hormones are taken into account. We have updated the section in the discussion to summarize these considerations in light of the new results (see pages 20-21, section: "A uniform molecular response to stress and noradrenaline release in both sexes").

Comment 3: A major finding of the present study is the involvement of noradrenergic transcriptomic changes occurring in astrocytic genes in the hippocampus. Given the stated importance of this finding within the discussion, it seems that some additional dialogue integrating this with current literature about the role of astrocytes in the hippocampus during stress or fear memory would be important.

Response: We thank the reviewer for giving us an opportunity to add a more detailed discussion about the role of astrocytes and thyroid hormones in the hippocampus during learning and memory formation. We have added these statements to the discussion:

“Within the hippocampus, astrocytic pathways are emerging as important players for learning and memory processes (Gibbs, Hutchinson, and Hertz 2008; Bohmbach et al. 2022). In fact, it is well-known that NA enhances memory consolidation (Schwabe et al. 2022; McGaugh and Roozendaal 2002), and recent work suggests that these effects are mediated by astrocytic β-adrenergic receptors (Gao et al. 2016; Iqbal et al. 2023). Our transcriptomic screens revealed Dio2 as the most prominent target influenced by LC activity. Dio2 is selectively expressed in astrocytes and encodes for the intracellular type II iodothyronine deiodinase, which converts thyroxine (T4) to the bioactive thyroid hormone 3,3',5-triiodothyronine (T3) and therefore regulates the local availability of T3 in the brain (Bianco et al. 2019). Enzymatic activity of DIO2 has further been shown to be increased by prolonged noradrenergic transmission through desipramine treatment in LC projection areas (Campos-Barros et al. 1994). This suggests that the LC-NA system and its widespread projections could act as a major regulator of brain-derived T3. Notably, T3-signaling plays a role in hippocampal memory formation (Rivas and Naranjo 2007; Sui et al. 2006), raising the possibility that NA-induced Dio2 activity in astrocytes might mediate some of these effects.”

Comment 4: The comparison of the candidate genes activated by the LC in the present study (swim) with datasets published by Floriou-Servou et al., 2018 (Novelty, swim, restraint, and footshock) is an interesting and important comparison. Were there other stressors identified in this paper or other publications that do not regulate these candidate genes? Further, can references be added to clarify to the reader, that prior studies have identified that novelty, restraint and footshock all activate LC-NA neurons.

ponse: Thank you for the positive feedback. We have only tested the stressors reported in Figure 4a-b (novelty, swim, restraint, and footshock). It is known that all these stressors trigger noradrenaline release, in fact we are not aware of stressors that do not trigger NA release. This reproducible finding supports the notion that the identified set of genes is indeed highly NAresponsive. As suggested, we have now included references that show increased NA release in response to all these stressors:

“Therefore, we assessed their expression in a dataset comparing the effect of various stressors on the hippocampal transcriptome (Floriou-Servou et al., 2018). The stressors included restraint, novelty and footshock stress, which have all previously been shown to increase hippocampal NA release (HajósKorcsok et al., 2003; Lima et al., 2019; Masatoshi Tanaka et al., 1982).”

Comment 5: Comparisons are made between chemogenetic studies and yohimbine, stating that fewer genes were activated by chemogenetic activation of LC neurons. There is clear justification for why this may occur, but a caveat may need to be mentioned, that evidence of neuronal activation in the LC by each of these methods were conducted at 90 (yohimbine) versus 45 (hM3Dq) minutes, and therefore it cannot be ruled out that differences in LC-NA activity levels might also contribute.

Response: The reviewer raises an important point about some inconsistencies between the time points chosen in our study, an aspect that was also pointed out by Reviewer #2. We have chosen the 45 and 90 min time points for two different reasons. On the one hand, cFos changes on the protein level are known to peak 90 min after neuronal activation, and we wanted to capture the strongest possible cFos signal in the LC. On the other hand, we wanted to measure gene expression changes triggered by NA release, which already occur 45 min after noradrenergic activation (Roszkowski et al., 2016). Thus, when the experimental design allowed separate experiments (e.g. systemic yohimbine injection), we chose to measure gene expression after 45 min, but to validate cFos activation in the LC separately after 90min. In response to DREADD activation, however, we wanted to confirm within the same animal that LC activation was successful, and thus we collected LC and hippocampus simultaneously (Figure 2c,d). While the cFos increase is already very pronounced at the 45min time point (Figure 2g), the quality of IHC is slightly lower because the tissue cannot be perfused in this experimental design. Therefore, we do not think that the time point for cFos sampling matters in this context. However, we agree with the reviewer that it remains unclear whether yohimbine and DREADDs activate the LC with similar potency. To directly compare NA release would require a set of photometry-based experiments to measure NA release using genetically-encoded NA-sensors. While we have added such experiments for LC activation with DREADDs and optogenetics to show rapid NA release indeed occurs in the hippocampus (see Reviewer #2, Comment 3; Supplementary Figure 2c-h), yohimbine interferes with the NA-sensors as explained in detail in response to Reviewer 2, Comment 3. Thus, it was too challenging for us to directly compare the release dynamics in response to DREADDs and yohimbine, which was also not the main focus of our work. To explicitly address this caveat, we have extended the corresponding section in the discussion:

"Finally, our observation that systemic administration of the α2-adrenergic receptor antagonist yohimbine very closely recapitulates the transcriptional response to stress stands in contrast to the much more selective transcriptional changes observed after chemogenetic or optogenetic LC-NA activation. This difference could be due to various factors. First, it remains unclear how strong the LC gets activated by yohimbine versus hM3Dq-DREADDs. However, given the potent LC activation observed after DREADD activation, it seems unlikely that yohimbine would lead to a more pronounced LC activation, thus explaining the stronger transcriptional effects. Second, contrary to LC-specific DREADD-activation, systemic yohimbine injection will also antagonize postsynaptic α2-adrenergic receptors throughout the brain (and periphery). More research is needed to determine whether this could have a more widespread impact on the hippocampus (and other brain regions) than isolated LC-NA activation, further enhancing excitability by preventing α2-mediated inhibition of cAMP production. Finally, systemic yohimbine administration and noradrenergic activity have been shown to induce corticosterone release into the blood (Johnston, Baldwin, and File 1988; Leibowitz et al. 1988; Fink 2016). Thus, yohimbine injection could have broader transcriptional consequences, including corticosteroid-mediated effects on gene expression."

Comment 6: Please add information about how virus or cannula placement was confirmed in these studies. Were missed placements also analyzed separately?

Response: Pupillometry recordings were performed with all animals involving optogenetic or chemogenetic manipulations of the LC, before subjecting them to stress experiments. These assessments account for both correct optic fiber placement and virus expression (Privitera et al., 2020). If an animal did not show a clear pupil response, it was not included any further in the study. To demonstrate correct cannula placement for drug infusion of isoprotenerol in the dorsal hippocampus, we added a representative image of cannula placement in Supplementary Figure 1h.

Comment 7: Time of day for tissue collection used in genetic analysis should be reported for all studies conducted or reanalyzed.

Response: Thank you for pointing out this omission. Tissue collection for RNA-seq analysis was always performed between 11am and 5pm during the dark phase of the reversed light-dark cycle. We have added this information to the corresponding method section (“Tissue collection”).

**Reviewer #1 (Recommendations For The Authors):**
Comment 8: This is a well written, comprehensive and rigorous manuscript that will be of great interest to those in the scientific community.

Response: Thank you for the positive evaluation of our work and for the constructive feedback.

**Reviewer #2 (Public Review):**
Comment 1: The present manuscript investigates the implication of locus coeruleus-noradrenaline system in the stress-induced transcriptional changes of dorsal and ventral hippocampus, combining pharmacological, chemogenetic, and optogenetic techniques. Authors have revealed that stress-induced release of noradrenaline from locus coeruleus plays a modulatory role in the expression of a large scale of genes in both ventral and dorsal hippocampus through activation of β-adrenoreceptors. Similar transcriptional responses were observed after optogenetic and chemogenetic stimulation of locus coeruleus. Among all the genes analysed, authors identified the most affected ones in response to locus coeruleus-noradrenaline stimulation as being Dio2, Ppp1r3c, Ppp1r3g, Sik1, and Nr4a1. By comparing their transcriptomic data with publicly available datasets, authors revealed that these genes were upregulated upon exposure to different stressors. Additionally, authors found that upregulation of Ppp1r3c, Ppp1r3g, and Dio2 genes following swim stress was sustained from 90 min up to 2-4 hours after stress and that it was predominantly restricted to hippocampal astrocytes, while Sik1 and Nr4a1 genes showed a broader cellular expression and a sharp rise and fall in expression, within 90 min of stress onset.Overall, the paper is well written and provides a useful inventory of dorsal and ventral hippocampal gene expression upregulated by activation of LC-NA system, which can be used as starting point for more functional studies related to the effects of stress-induced physiological and pathological changes.

Response: We thank the reviewer for the careful assessment of our work.

Comment 2: However, I believe that the study would have benefited of a more comprehensive analyses of sex differences. Experiments in females were conducted only in one experiment and analyses restricted to the ventral hippocampus.

Response: In response to the comments by the reviewer, as well as Reviewer #1 and the editors, we have sequenced an additional 32 brain samples to expand the comparison of sex effects in females and males across dorsal and ventral hippocampus, and we included a new meta-analysis of 3 experimental datasets (51 male and 20 female) samples, to thoroughly assess sex differences in the transcriptomic response to stress. We refer the reviewer to our detailed response provided above to Reviewer #1, comment #2, and the updated results section on pages 3-4.

Comment 3: Although, the experiments were overall sound and the results broadly support the conclusion made, I think some methodological choices should be better explained and rationalized. For instance, the study focuses on identifying transcriptional changes in the hippocampus induced by stress-mediated activation of the LC-NA system, however NA release following stress exposure and pharmacological or optogenetic manipulation was mostly measured in the cortex.

Response: Because the hippocampus was used for RNA-sequencing, we could not assess NA release in the hippocampus (as this would require fiber implants that would interfere with molecular measures, or different tissue processing for HPLC). Nonetheless, we wanted to assess the transcriptional changes in the hippocampus, while simultaneously measuring successful stimulation of the LC-NA system in the same animals. To achieve this, we pursued 3 routes: (1) we used pupillometry to confirm functional LC activation; (2) we measured cFOS in the LC to directly demonstrate LC activation; (3) we assessed NA release using uHPLC (which requires larger tissue samples) and we chose the cortex because both cortex and hippocampus receive NA predominantly from the LC (Samuels & Szabadi, 2008). Importantly, we had previously shown that chemogenetic LC activation leads to a similar NA turnover in both the cortex and hippocampus, as measured by uHPLC (Zerbi et al., 2019). The relevant figure from that paper is inserted below to quickly show the striking similarity between hippocampus and cortex.

**Author response image 1. sa2fig1:** Levels of noradrenaline (NE) turnover (MHPG/NE ratio) in the cortex (CTX) and hippocampus (HC), measured in whole tissue with uHPLC 90min after hM3Dq-DREADD activation of the LC (copied and cropped from Zerbi et al, 2019, Neuron).

In response to the reviewers comment, we performed additional experiments to directly demonstrate that LC-activation with DREADDs as well as optogenetics causes an increase in hippocampal NA-release. We recorded NA release in the hippocampus (using fiber photometry combined with genetically encoded NA sensors). For DREADD activation, we observed a strong increase in hippocampal noradrenaline that started a few minutes after clozapine administration, and this increase was sustained throughout the duration of the 21 minute recording (see Supplementary Figure2c-e). For optogenetic LC activation, we find a rapid and immediate sharp increase in NA levels in the hippocampus (Supplementary Figure 2f-h). These experiments were performed in females and males and triggered similar responses. An adapted and cropped version of Supplementary Figure 2 is pasted below for ease of reading.

Please note that we could not perform a similar experiment using yohimbine, because the GRABNE sensors are based on the alpha-2 adrenergic receptor, thus yohimbine administration interferes with the photometry recording. However, we believe that it is clear from this response that strong activation of the LC leads to uniform release of NA in the hippocampus and cortex.

**Author response image 2. sa2fig2:** c, Schematic of fiber photometry recording of hippocampal NA during chemogenetic activation of the LC. After 5 min baseline recording in the homecage animals were injected with clozapine (0.03mg/kg, i.p.) and placed in the OFT for 21min. d, Average ΔF/F traces of GRABNE2m photometry recordings in response to chemogenetic activation of the LC (mean ± SEM for hM3DGq+ and hM3DGq- split into females and males, n=3/group/sex). e, Peak ΔF/F response of fiber photometry trace. f, Schematic of fiber photometry recording of hippocampal NA during optogenetic activation of the LC. Animals were lightly anesthetized (1.5% isoflurane) and recorded in a stereotaxic frame. After 1 min baseline recording, animals were stimulated three times with 5Hz for 10s (10ms pulse width, ~8mW laser power) and recorded for 2 min post-stimulation. g, Average ΔF/F traces of the NA sensors GRABNE1m and nLightG in response to optogenetic activation of the LC mean ± SEM for females and males, n(females) = 10, n(males)=5. h, Peak ΔF/F response of fiber photometry trace.

Comment 4: Furthermore, behavioral changes following systemic pharmacologic or chemogenetic manipulation were observed in the open field task immediately after peripheral injections of yohimbine or CNO, respectively. Is this timing sufficient for both drugs to cross the blood brain barrier and to exert behavioral effects?

Response: We have previously shown that chemogenetic activation of the LC through clozapine elicits pupil responses within 1-2 minutes after injection (Privitera et al., 2020; Zerbi et al., 2019). This indicates that clozapine rapidly crosses the blood brain barrier and affects LC activity within a few minutes after injection. Our additional experiments using genetically encoded sensors in the hippocampus show this even more directly (Supplementary Figure 2d), see also the response to Comment 3 above.

Similarly, yohimbine also rapidly crosses the blood brain barrier within the same time frame (Hubbard et al., 1988). These observations are consistent with the rapid behavioral effects that can be detected within a few minutes after injection of clozapine for LC-DREADD activation (Zerbi et al., 2019), and for yohimbine as well (von Ziegler et al., 2023). In response to another comment of this reviewer, we have also re-analyzed the behavior presented in the current manuscript in time-bins of 3 minutes, which also shows the rapid onset of effects in response to yohimbine (within the first 3 min) and DREADDs (within 6 min), see Supplementary Fig. 3.

Comment 5: Finally, the study shows that activation of noradrenergic hippocampus-projecting LC neurons is sufficient to regulate the expression of several hippocampal genes, although the necessity of these projection to induce the observed transcriptional effects has been tested to some extent through systemic blockade of beta-adrenoceptor, I believe the study would have benefited of more selective (optogenetic or chemogenetic) necessity experiments.

Response: We understand the reviewer's point that blocking the LC during stress exposure would be an interesting experiment. However, it is very hard to completely silence the LC during intense stressors. In fact, despite intense efforts, we have not been able to silence the LC during swim stress exposure using DREADDs or other chemogenetic approaches (PSAM/PSEM). We were in fact able to silence the LC with the optogenetic inhibitor JAWS (and others have reported successful LC silencing with GtACR2), but there is a major issue involving the "rebound effect", where more NA is released once the inhibition is stopped. We would thus have had to optogenetically silence the LC for 45-90 min, which would create heat artifacts, and require challenging control experiments to draw firm conclusions. Given all these issues, we reasoned that blocking adrenergic receptors is a simple and elegant solution, which provides clear evidence for the necessity of beta-adrenergic signaling.

**Reviewer #2 (Recommendations For The Authors):**
Major concerns:Comment 6: The study focuses on the identification of transcriptional changes in the hippocampus induced by stress-mediated activation of the LC-NA system, however, noradrenaline release following stress exposure or yohimbine injection was measured in the cortex. Authors should consider measuring NA concentrations in the hippocampus after exposure to swim stress or administration of yohimbine, or at least explain their choice to analyse to cortex in the manuscript.

Response: We have addressed this issue in detail in Response to "Reviewer 2, Comment #3", where we provided an overview of the additional data that support our approach. As mentioned before, measuring NA release after yohimbine is not compatible with our GRABNE-photometry approach, as the GRAB-sensor is based on alpha2-adrenoceptor. Here, we would like to add that measuring NA release using photometry during swim stress is also challenging. The challenge is the vigorous movement (swimming, typically in one direction), which creates pressure on the cables/implants. We felt that overcoming these experimental challenges (setup, troubleshooting and controls) would be beyond the scope of the paper, given that it is already known that this stressor leads to strong NA release in the hippocampus. We have now included references that demonstrate that all the stressors used in our work trigger NA increase in the hippocampus (see response to Reviewer 1, Comment 3): “Therefore, we assessed their expression in a dataset comparing the effect of various stressors on the hippocampal transcriptome (Floriou-Servou et al., 2018). The stressors included restraint, novelty and footshock stress, which have all previously been shown to increase hippocampal NA release (Hajós-Korcsok et al., 2003; Lima et al., 2019; Masatoshi Tanaka et al., 1982).”

Comment 7: Concerning the experiment aimed at investigating sex differences in gene expression, it is not clear the reason why authors decided to restrict their analyses in females to the ventral hippocampal only. The explanation that in males they did not detect major differences between the dorsal and ventral hippocampus is not sufficient, because there could have been different effects in females. Therefore, the conclusion made by the authors that their "results suggest that the transcriptomic response is independent of sex" is not entirely correct, since sex differences were only evaluated in the ventral hippocampus.

Response: We appreciate the reviewer's critique. As described above, we have now also sequenced the dorsal hippocampal tissue from the propranolol experiment (males and females, 32 samples) and additionally added an extensive meta-analysis of three large datasets (n=71) to compare transcriptional sex differences in response to stress. A detailed description of these experiments and how they have extended/supported our conclusions have been provided in response to Reviewer #1, Comment #2.

Comment 8: Besides the effects on females, the same experiment examined whether propranolol by itself (in the absence of stress) would have been able to alter gene expression: such effects were not examined in the dorsal hippocampus. In contrast, in a different experiment, the effects of isoproterenol on genes expression were restricted to the dorsal hippocampus only. Furthermore, related to this latter experiment, intra-dorsal hippocampal injection of isoproterenol should presumably mimic the rise in NA observed after stress exposure, why was gene expression measured 90 min after isoproterenol central injections while in the other experiments gene expression was determined 45 min after stress, that is when authors observe the peak NA concentration?

Response: We have addressed the reviewer's critique of dorsal vs ventral hippocampus by reanalyzing 32 additional samples from dorsal hippocampus of male and female mice after propranolol (or saline) injection. Please see response to Reviewer #1, comment #2.

Regarding the time points: We have chosen the 45 and 90 min time points mainly for two reasons. First, cFos protein changes are known to be strongest 90 min after neuronal activation. Second, because we wanted to capture gene expression changes triggered by NA release, we reasoned that these effects must be fast and should thus be measured at an early transcriptional time-point (45min). However, after performing the time-course experiment after swim stress exposure (Figure 4d,c), we observed that the LC-NA-sensitive genes (e.g. Dio2 and several PP1-subunits) show the strongest changes 90 min after stress exposure. Therefore, in some of our experiments we opted to analyze gene expression changes at 90min, converging with the time-point we typically use for cFos staining. Contrary to the reviewer's statement, peak NA concentrations are not observed 45 min after the various interventions, but rather the peak in the main metabolite (MHPG) is observed then, due to the temporal dynamics of NA release and breakdown. NA release occurs immediately upon stress exposure (or direct LC activation), which we also show in the new photometry data described above. Thus, rapid NA release triggers intracellular cascades that lead to downstream transcriptional changes, which peak presumably between 4590 min later.

Comment 9: Behavioral changes following systemic pharmacologic or chemogenetic manipulation were observed in the open field task immediately after peripheral injections of yohimbine or CNO, respectively. Is this timing sufficient for both drugs to cross the blood brain barrier and to exert behavioral effects? It is also not immediately clear the reason why the open field tasks have different durations depending on the experiments, which can also impact the results. Authors might also consider to split the open field data analyses in 2 or 3 min time-bins, to allow for a better comparison across the different results.

Response: We thank the reviewer for the suggestion to plot the behavior data as time-bins. We have implemented this change for the yohimbine and DREADD experiments, and updated the corresponding figure accordingly (Supplementary Figure 3, pasted below for ease of reading). The new visualization clearly shows that yohimbine injection triggers rapid behavioral effects already in the first three minutes, whereas the LC-DREADD activation triggers behavioral changes within 3-6 minutes after injection. Thus, clear drug effects are visible in the first 10 minutes, which is comparable to the standard OFT test (10min testing) shown in response to swim stress exposure (Suppl. Figure 3a). The choice to expose mice to the OFT for 21 minutes in total was due to the fact that we based our experimental approach on the optogenetic LC-stimulation protocol first published by McCall and colleagues (McCall et al, Neuron, 2015), in which the LC is stimulated for 3 min followed by 3 min pauses (see Suppl. Figure 3d). Because of this on-off design, we decided to keep the optogenetic analysis simple and show the overall effect (Supplementary Figure 3d), particularly as we know that NA dynamics do not recover rapidly enough after 3 min continuous stimulation to justify a bin-analysis (unpublished data).

**Author response image 3. sa2fig3:** Effects of acute stress and noradrenergic stimulation on anxiety-like behaviour in the open field test. a, Stress-induced changes in the open field test 45 min after stress onset. Stressed animals show overall reductions in distance traveled (unpaired t-test; t=3.55, df=22, p=0.0018), time in center (welch unpaired t-test; t=3.50, df=13.61, p=0.0036), supported rears (unpaired t-test; t=3.39, df=22, p=0.0026) and unsupported rears (unpaired t-test; t=5.53, df=22, p = 1.47e-05) compared to controls (Control n = 12; Stress n = 12). This data have been previously published (von Ziegler et al., 2022). b, Yohimbine (3 mg/kg, i.p.) injected animals show reduced distance traveled (unpaired t-test; t=2.39, df=10, p=0.03772), reduced supported rears (unpaired t-test; t=6.56, df=10, p=0.00006) and reduced unsupported rears (welch unpaired t-test; t=3.69, df=4.4, p = 0.01785) compared to vehicle injected animals (Vehicle n = 6; Yohimbine n = 7). c, Chemogenetic LC activation induced changes in the open field test immediately after clozapine (0.03 mg/kg, i.p.) injection. hM3Dq+ animals show reduced distance traveled (unpaired t-test; t=6.28, df=13, p=0.00003), reduced supported rears (unpaired t-test; t=4.28, df=13, p=0.0009), as well as reduced unsupported rears (welch unpaired t-test; t=4.28, df=13, p = 0.00437) compared to hM3D- animals (hM3Dq- n = 7; hM3Dq+ n = 8). d, Optogenetic 5 Hz LC activation induced changes during the open field test. ChR2+ animals show reduced supported rears (unpaired t-test; t=2.42, df=64, p=0.0185) and reduced unsupported rears (unpaired ttest; t=2.91, df=64, p = 0.00499) compared to ChR2- animals (ChR2- n = 32; ChR2+ n = 36). Data expressed as mean ± SEM. *p < 0.05, **p < 0.01, ***p < 0.001, ****p < 0.0001.

Comment 9: The study shows that activation of noradrenergic hippocampus-projecting LC neurons is sufficient to regulate the expression of several hippocampal genes. I believe the study would have benefited of more selective necessity experiments. Authors might consider adding optogenetic (or chemogenetic) experiments aimed at inhibiting LC-NA hippocampal projections during stress exposure (or, alternatively, perform intrahippocampal pharmacological blockade of β-adrenoreceptors during stress exposure), and determine the effects on gene expression.

Response: We kindly refer the reviewer to our previous response to Comment #2 above.

Minor concerns:There is a typo in the abstract. Please correct "LN-NA" with "LC-NA"

Response: Thank you, we have corrected it.

References

Bangasser, D. A., Eck, S. R., & Ordoñes Sanchez, E. (1/2019). Sex differences in stress reactivity in arousal and attention systems. Neuropsychopharmacology: Official Publication of the American College of Neuropsychopharmacology, 44(1), 129–139.

Bangasser, D. A., Wiersielis, K. R., & Khantsis, S. (06/2016). Sex differences in the locus coeruleusnorepinephrine system and its regulation by stress. Brain Research, 1641, 177–188.